# URS: A Unified Neural Routing Solver for Cross-Problem Zero-Shot Generalization

Changliang Zhou [1 2]  Canhong Yu [3]  Shunyu Yao [4]  Xi Lin [5]  Zhenkun Wang [1 2]  Yu Zhou [3]  Qingfu Zhang [4]

## Abstract

Multi-task neural routing solvers have emerged as a promising paradigm for their ability to solve multiple vehicle routing problems (VRPs) using a single model. However, existing neural solvers typically rely on predefined problem constraints or require per-problem fine-tuning, which substantially limits their zero-shot generalization ability to unseen VRP variants. To address this critical bottleneck, we propose URS, a unified neural routing solver that achieves zero-shot generalization across a wide range of unseen VRPs with a single model. We propose a unified data representation (UDR) that replaces problem enumeration with data unification, thereby broadening the problem coverage and reducing reliance on domain expertise. In addition, we introduce a mixed bias module (MBM) during encoding to improve node embeddings, which efficiently captures multiple priors inherent to various problems. On top of the UDR, we develop a problem-conditioned parameter generator to further improve zero-shot generalization. Extensive experiments show that URS consistently produces high-quality solutions for 110 VRP variants (including 99 unseen variants) while demonstrating impressive scalability to large-scale instances with up to 7000 nodes. To the best of our knowledge, URS is the first neural solver to handle over 100 VRP variants with a single model. Our code is available at https://github.com/CIAM-Group/URS.

## 1. Introduction

The Vehicle Routing Problem (VRP) is an essential class of combinatorial optimization problems (COPs) with extensive applications in logistics and supply chain management (Tiwari & Sharma, 2023; Sar & Ghadimi, 2023). Solving VRPs efficiently is challenging due to their NP-hard nature. While exact solvers can produce optimal solutions, they often become computationally prohibitive for real-world scenarios. In recent decades, classical heuristic solvers (Helsgaun, 2017; Vidal, 2022) have achieved impressive performance within acceptable timeframes. Nevertheless, these methods require considerable domain expertise to design specialist rules for each routing problem. Given the growing diversity of VRP variants in real-world applications, manually crafting tailored rules for every case has become impractical.

In recent years, neural combinatorial optimization (NCO) methods have attracted substantial attention for their potential to reduce reliance on handcrafted rules while maintaining competitive solution quality (Bengio et al., 2021; Ba et al., 2026). This has led to the development of many high-performing neural routing solvers that automatically learn implicit problem-specific rules from data under different training paradigms such as supervised learning (SL) (Drakulic et al., 2023; Luo et al., 2023; 2025b), reinforcement learning (RL) (Kool et al., 2019; Zhou et al., 2026), and self-improved learning (SIL) (Luo et al., 2025a; Pirnay & Grimm, 2024). Although these solvers have demonstrated impressive performance on specific problems (Huang et al., 2025; Zhou et al., 2025), they typically require architectural customization and per-problem retraining to accommodate the distinct constraints and features of different VRP variants, thereby increasing overall training costs and hindering practical deployment to new problems.

To address the challenge of cross-problem generalization, growing attention has been directed toward multi-task learning capable of handling diverse routing problems. As summarized in Table 1, existing efforts largely fall into two categories: 1) constraint combination-based multi-task learning and 2) adapter-based fine-tuning. In the first approach, VRP variants are treated as different combinations of constraint attributes, and a unified model is trained across these combinations to enable knowledge sharing for problems with

---

[1]School of Automation and Intelligent Manufacturing, Southern University of Science and Technology, Shenzhen, China [2]Guangdong Provincial Key Laboratory of Fully Actuated System Control Theory and Technology, Southern University of Science and Technology, Shenzhen, China [3]College of Computer Science and Software Engineering, Shenzhen University, Shenzhen, China [4]Department of Computer Science, City University of Hong Kong, Hong Kong SAR, China [5]School of Mathematics and Statistics, Xi'an Jiaotong University, Xi'an, China. Correspondence to: Zhenkun Wang <wangzk3@sustech.edu.cn>.

*Proceedings of the 43rd International Conference on Machine Learning*, Seoul, South Korea. PMLR 306, 2026. Copyright 2026 by the author(s).

*Table 1.* Comparison between our URS and existing neural solvers with multi-task learning. Note that "#VRP Variants" and "Generalizable Scale" refer to the total number of tested problems and the maximum scale reported in the original papers, respectively.

| Multi-task Neural Routing Solver | Training Paradigm | #VRP Variants | Symmetric VRPs | Asymmetric VRPs | Pickup and Delivery Problems | Zero-shot Generalization | Training Scale | Generalizable Scale | Remarks |
|---|---|---|---|---|---|---|---|---|---|
| MTPOMO (Liu et al., 2024) | RL | 16 | ✓ | × | × | ✓ | 100 | 1000 | Constraint Combination |
| MVMoE (Zhou et al., 2024) | RL | 16 | ✓ | × | × | ✓ | 100 | 1000 | Constraint Combination |
| RouteFinder (Berto et al., 2025) | RL | 48 | ✓ | × | × | ✓ | 100 | 1000 | Constraint Combination |
| CaDA (Li et al., 2025a) | RL | 16 | ✓ | × | × | ✓ | 100 | 200 | Constraint Combination |
| MTL-KD (Zheng et al., 2025) | RL+KD | 16 | ✓ | × | × | ✓ | 100 | 1000 | Constraint Combination |
| TSP-FT (Lin et al., 2024) | RL | 4 | ✓ | × | × | × | 100 | 1000 | Adapter-based Fine-tuning |
| MTL-MAB (Wang et al., 2025a) | RL | 3 | ✓ | × | × | × | 100 | 1000 | Adapter-based Fine-tuning |
| GOAL (Drakulic et al., 2025) | SL | 10 | ✓ | ✓ | × | × | 100 | 1000 | Adapter-based Fine-tuning |
| **URS (Ours)** | **RL** | **110** | ✓ | ✓ | ✓ | ✓ | **100** | **7000** | **Unified Data Representation** |

seen constraints (Liu et al., 2024; Zhou et al., 2024; Li et al., 2025a; Berto et al., 2025; Zheng et al., 2025). The second approach builds a shared model backbone for all problems and incorporates problem-specific input/output adapters, thereby reducing retraining costs (Lin et al., 2024; Drakulic et al., 2025; Wang et al., 2025a).

Despite the above advancements, existing methods still fall short in cross-problem zero-shot generalization. The problem coverage of constraint combination-based approaches is inherently bounded by their manually specified constraint sets, whereas adapter-based approaches require additional fine-tuning and thus cannot perform zero-shot generalization. Furthermore, both strategies fundamentally rely on the explicit problem enumeration through a predefined set of problem tags. This practice is challenging because the constraint space of VRP variants is open-ended and compositional. More critically, creating and maintaining such a problem taxonomy requires considerable domain expertise, which NCO always aims to avoid. A detailed review of related work on single-task and multi-task learning is provided in Appendix A.

In this paper, we propose a powerful **U**nified Neural **R**outing **S**olver (URS) to improve the cross-problem zero-shot generalization ability for NCO methods. Our contributions can be summarized as follows: (1) We propose a unified data representation (UDR) that replaces existing problem enumeration with data unification, significantly broadening problem coverage while mitigating reliance on domain-specific expertise; (2) We introduce a mixed bias module (MBM) during encoding to improve node embeddings, which efficiently captures multiple priors inherent to various problems while reducing architectural redundancy; (3) On top of the UDR, we develop a problem-conditioned parameter generator to further improve zero-shot generalization; (4) Extensive experiments on **110** VRP variants show that URS not only achieves competitive performance against specialist neural solvers on seen variants but also exhibits strong zero-shot generalization across **99** unseen variants. Notably, URS demonstrates impressive scalability to large-scale instances with up to 7000 nodes. To the best of our knowledge, URS is the first neural solver to efficiently solve over 100 VRP variants with a single model, without retraining or fine-tuning.

## 2. Preliminaries

In this section, we first introduce the definition of VRP, then provide an overview of recent constructive neural solvers for solution generation (Kool et al., 2019; Kwon et al., 2020).

**Vehicle Routing Problems** Consider a VRP instance with an optional depot indexed by $0$ and $n$ customers indexed by $\{1, 2, \ldots, n\}$, such as in the Capacitated VRP (CVRP). The instance can be represented as a graph $\mathcal{G} = (\mathcal{V}, \mathcal{E})$, where the node set $\mathcal{V} = \{v_i\}_{i=0}^n$ has a total size of $|\mathcal{V}| = 1 + n$ unless the variant has no depot (e.g., the Traveling Salesman Problem), and the edge set is $\mathcal{E} = \{e(v_i, v_j) \mid v_i, v_j \in \mathcal{V}, v_i \neq v_j\}$. The travel cost between nodes is given by a distance matrix $\boldsymbol{D} = \{d_{ij} \mid \forall i, j \in 0, \ldots, n\}$. In VRP, each node $v_i \in \mathcal{V}$ includes node coordinates $\{x_i, y_i\}$ when available and problem-specific attributes (e.g., demands in CVRP). Formally, a feasible solution is denoted by a node permutation $\boldsymbol{\pi} = (\pi_1, \pi_2, \ldots, \pi_m)$ (Toth & Vigo, 2002) (i.e., a finite sequence) that represents a set of valid routes. Each route is traversed by a single vehicle originating and terminating at its own depot or starting node, while strictly satisfying all constraints. Let $\Omega$ denote the feasible sequence set, given an objective function $f(\boldsymbol{\pi}|\mathcal{G})$, we aim to search for a sequence $\boldsymbol{\pi}^*$ with the maximum objective, i.e., $\boldsymbol{\pi}^* = \arg\max_{\boldsymbol{\pi} \in \Omega} f(\boldsymbol{\pi}|\mathcal{G})$. In most VRPs, $f(\boldsymbol{\pi}|\mathcal{G})$ can be defined as the negative value of the total distance of $\boldsymbol{\pi}$.

In this paper, we consider 110 VRP variants which may simultaneously have one or more constraints from the following categories: (1) Capacity (C); (2) Open Route (O); (3) Backhaul (B); (4) Backhaul and Priority (BP); (5) Duration Limit (L); (6) Time Windows (TW); (7) Multi-Depot (MD); (8) Prize Collecting (PC); (9) Asymmetric (A); and (10) Pickup and Delivery (PD). Detailed definitions of these constraints are provided in Appendix B.

**Constructive Neural Routing Solver** Most constructive NCO methods employ an encoder-decoder architecture for solution construction (Luo et al., 2023; Kwon et al., 2020). Let learnable model parameters be denoted as $\boldsymbol{\theta} = \{\boldsymbol{\theta}_{enc}, \boldsymbol{\theta}_{dec}\}$. Without loss of generality, we present the prevailing autoregressive construction pipeline using AM (Kool et al., 2019). Given an instance $\mathcal{G}$, raw node

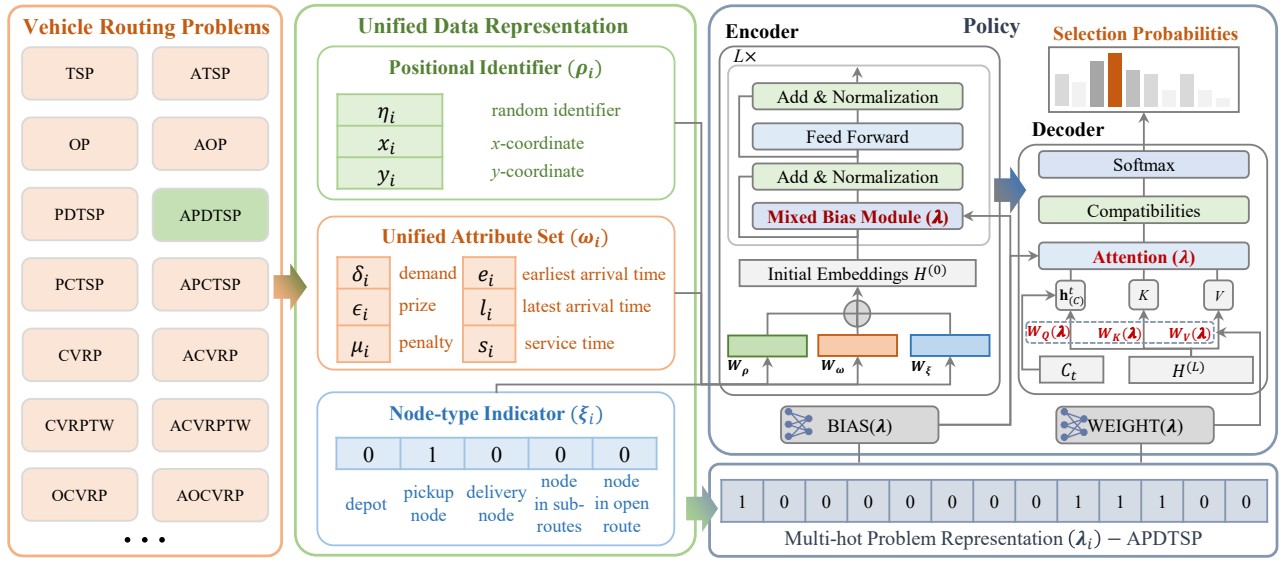

*Figure 1.* The pipeline of our URS for solving 110 VRP variants using a single model without any fine-tuning, illustrated on APDTSP.

features are first mapped by a linear projection to initial embeddings $H^{(0)} = \{\mathbf{h}_i^{(0)}\}_{i=0}^{n}$. They are then processed by an encoder $\boldsymbol{\theta}_{enc}$ with $L$ attention layers to produce refined node embeddings $H^{(L)} = \{\mathbf{h}_i^{(L)}\}_{i=0}^{n}$. At each decoding step $t$, the decoder $\boldsymbol{\theta}_{dec}$ sequentially selects a node to append to the current partial solution $\pi_{1:t-1} = (\pi_1, \pi_2, \ldots, \pi_{t-1})$, where $\pi_1, \pi_{t-1} \in \mathcal{V}$ are the **first** and **last** visited node, respectively. This sequential process continues until a complete solution is formed. To guarantee the feasibility of generated solutions, a masking function $\mathcal{M}_t$ sets the selection probabilities of the following nodes to $-\infty$ during the construction process: nodes that (1) are visited or (2) violate problem-specific constraints (e.g., capacity).

## 3. Methodology

In this section, as illustrated in Figure 1, we propose a powerful **U**nified Neural **R**outing **S**olver (URS), which significantly improves cross-problem zero-shot generalization for NCO methods. Its key components are elaborated below.

### 3.1. Unified Data Representation

Existing multi-task neural solvers (Zhou et al., 2024; Drakulic et al., 2025) fundamentally rely on explicit problem enumeration via a predefined set of problem tags, which is challenging because the constraint space of VRPs is open-ended and compositional. More critically, creating and maintaining such a problem taxonomy requires considerable domain expertise, which NCO always aims to avoid.

From a data perspective, despite the substantial diversity in constraints, their instances share a common structural representation. For existing constraint combination-based meth-

ods (Liu et al., 2024; Zhou et al., 2024), the problems they evaluate can be formulated as CVRPTW instances. Leveraging this structural commonality, we propose a unified data representation (UDR) $\boldsymbol{U} = \{\mathbf{u}_i\}_{i=0}^{n}$. By replacing explicit problem enumeration with data unification, our UDR broadens problem coverage and reduces reliance on domain expertise. Notably, the inputs to existing multi-task neural solvers based on either problem-specific adapters (Drakulic et al., 2025) or constraint combinations (Zhou et al., 2024) can be viewed as special subsets of our proposed UDR.

Instead of relying on a predefined problem taxonomy with a discrete set of problem tags, UDR decouples instance representation from constraint definitions and operates in a unified space characterized by both continuous and discrete features. It allows a single neural solver to address a much larger, open-ended space of VRP variants, as new problems can be seamlessly incorporated into this unified representation. Problem-specific constraints are delegated to the model-agnostic masking function $\mathcal{M}_t$. For example, for all CVRP variants, only the remaining load is explicitly provided as input, while diverse additional constraints (e.g., time windows) are enforced implicitly by masking infeasible candidate nodes during solution construction, rather than being fully enumerated in the decoder's input. Thus, UDR enables a single model to handle various VRP variants while significantly reducing reliance on domain expertise. Specifically, for an instance $\mathcal{G}$, each $\mathbf{u}_i = \{\boldsymbol{\rho}_i, \boldsymbol{\omega}_i, \boldsymbol{\xi}_i\}$, where $\boldsymbol{\rho}_i, \boldsymbol{\omega}_i, \boldsymbol{\xi}_i$ are a positional identifier, unified attribute set, and node-type indicator, respectively.

**Positional Identifier ($\boldsymbol{\rho}_i$)** We define a unified positional identifier for each node as $\boldsymbol{\rho}_i = \{\eta_i, x_i, y_i\} \in [0,1]^3$. The $\eta_i \sim \text{Uniform}(0,1) \in \mathbb{R}^1$ is a sampled scalar, which is

used to address asymmetric problems, following Drakulic et al. (2023). For symmetric cases, we simply set $\eta_i = 0$.

**Unified Attribute Set ($\boldsymbol{\omega}_i$)**  We define the unified attribute set as $\boldsymbol{\omega}_i = \{\delta_i, \epsilon_i, \mu_i, e_i, l_i, s_i\}$, where they correspond to demand, prize, penalty, earliest arrival time, latest arrival time, and service time, respectively. The unification empowers the model to learn the relative importance of individual attributes in a shared embedding space. It also supports effortless attribute extension or ablation via simple zero-filling, eliminating the need to change the architecture.

**Node-type Indicator ($\boldsymbol{\xi}_i$)**  We provide a 5-dimensional binary vector $\boldsymbol{\xi}_i \in \{0,1\}^5$ for each node, encoding: depot, pickup node, delivery node, node in sub-routes, node in open route. The nodes in sub-routes and open routes mean the solution $\pi$ can have sub-routes or be an open route. We introduce explicit structural roles to enable the model to generalize over diverse variants. Further ablation analysis concerning $\boldsymbol{\xi}_i$ is provided in Appendix K.1.

**Multi-hot Problem Representation ($\boldsymbol{\lambda}_i$)**  Through the UDR, each problem instance activates only a specific subset of the unified feature space. For any instance $\mathcal{G}_i$ drawn from different problems, a corresponding multi-hot problem representation $\boldsymbol{\lambda}_i$ is obtained by indicating active (non-zero) feature slots with a value of $1$. This representation captures only the presence of features, in contrast to their raw values, and it subsequently provides conditional guidance to the position bias weight (see Equation (6)) and the adaptive decoder parameters $\boldsymbol{\theta}_{dec}(\boldsymbol{\lambda})$ (see Equation (9) and Equation (10)), thereby helping the model in distinguishing problem variants and refining its node selection process.

A more detailed description of $U$ and $\boldsymbol{\lambda}$ is provided in Appendix C and Appendix D, respectively.

### 3.2. Model Architecture

As shown in Figure 1, we adopt AM (Kool et al., 2019) as our basic model. While UDR provides cross-problem consistency, different problems have diverse geometric priors. To efficiently learn the multiple priors inherent in various problems and obtain better cross-problem zero-shot generalization, we propose two key enhancements: (1) a mixed bias module (MBM) that captures multiple priors inherent to various problems, and (2) problem-conditioned parameter generators conditioned on the UDR. Their implementations are detailed below.

**Embedding Layer**  Given an instance $\mathcal{G}$, for each $\mathbf{u}_i = \{\boldsymbol{\rho}_i, \boldsymbol{\omega}_i, \boldsymbol{\xi}_i\}$, we project it into $d$-dimensional embeddings through three separate linear transformations:

$$\mathbf{h}_i^{(0)} = \boldsymbol{\rho}_i W_{\boldsymbol{\rho}} + \boldsymbol{\omega}_i W_{\boldsymbol{\omega}} + \boldsymbol{\xi}_i W_{\boldsymbol{\xi}}, \qquad (1)$$

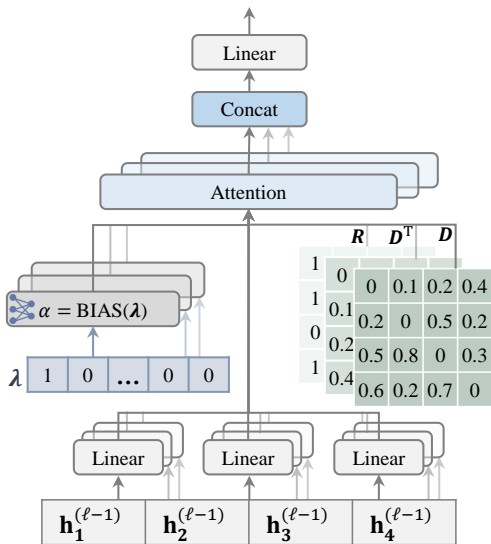

*Figure 2.* **The structure of the proposed mixed bias module, explained on a** $4$**-node APDTSP instance.** We replace plain attention with MBM during encoding to improve node embeddings, which effectively captures multiple priors inherent to various problems while reducing architectural redundancy.

where $W_{\boldsymbol{\rho}} \in \mathbb{R}^{3 \times d}$, $W_{\boldsymbol{\omega}} \in \mathbb{R}^{6 \times d}$, $W_{\boldsymbol{\xi}} \in \mathbb{R}^{5 \times d}$ are learnable matrices. Then we obtain a set of initial embeddings $H^{(0)} = \{\mathbf{h}_i^{(0)}\}_{i=0}^n$ for all nodes in instance $\mathcal{G}$. This embedding is the initial input of encoder $\boldsymbol{\theta}_{enc}$. After passing through $L$ stacked attention layers, $H^{(0)}$ is transformed into advanced node embeddings $H^{(L)} = \{\mathbf{h}_i^{(L)}\}_{i=0}^n$. The detailed encoding process is provided in Appendix E.1.

**Mixed Bias Module**  Given the distinct priors across VRP variants, efficiently capturing the intrinsic geometric bias is pivotal for enhancing the cross-problem generalization of RL-based multi-task neural solvers. Current leading RL-based models generally adopt a heavy encoder and light decoder architecture, where the quality of the encoder-generated node embeddings plays a key role in overall performance. While prior studies like MatNet (Kwon et al., 2021) have attempted to handle asymmetry by integrating directional distance biases via parallel layers, such designs incur high model complexity and lack universality. Consequently, effectively encoding multiple inherent biases, such as symmetric and asymmetric distances, and relational dependencies, remains challenging, hindering the acquisition of high-quality solutions across a wide range of problems.

To address the limitation, we propose the mixed bias module (MBM) to replace the vanilla attention mechanism (Vaswani et al., 2017), which enhances cross-problem generalization while reducing architectural redundancy. As shown in Figure 2, MBM can efficiently capture three problem-specific prior matrices via three separate attention calculations: (1) an outgoing distance matrix $\boldsymbol{D}$; (2) an incoming

distance matrix $\boldsymbol{D}^{\mathrm{T}}$; and (3) an optional relation matrix $\boldsymbol{R} = \{r_{ij} | \, \forall \, i, j \in 0, \ldots, n\}$, where $r_{ij} = 0$ if a predefined relation exists (e.g., pickup-delivery pairing) and 1 otherwise. This ensures that smaller values are preferred, aligning with the distance metric, where smaller values correspond to a larger selection bias. For the $\ell$-th layer, the MBM output $\hat{\mathbf{h}}_i^{(\ell)}$ can be expressed as:

$$\bar{\mathbf{h}}_i^{(0)} = \mathrm{Attention}(\mathbf{h}_i^{(\ell-1)}, H^{(\ell-1)}, f\left(\alpha, N, \boldsymbol{D}_i\right)), \quad (2)$$

$$\bar{\mathbf{h}}_i^{(1)} = \mathrm{Attention}(\mathbf{h}_i^{(\ell-1)}, H^{(\ell-1)}, f\left(\alpha, N, \boldsymbol{D}_i^{\mathrm{T}}\right)), \quad (3)$$

$$\bar{\mathbf{h}}_i^{(2)} = \begin{cases} \mathrm{Attention}(\mathbf{h}_i^{(\ell-1)}, H^{(\ell-1)}, f\left(\alpha, \boldsymbol{R}_i\right)) & \text{if } \boldsymbol{R} \neq \emptyset \\ \mathbf{0} & \text{otherwise,} \end{cases} \quad (4)$$

$$\hat{\mathbf{h}}_i^{(\ell)} = \left[\bar{\mathbf{h}}_i^{(0)}, \bar{\mathbf{h}}_i^{(1)}, \bar{\mathbf{h}}_i^{(2)}\right] W^O, \quad (5)$$

where $[\cdot, \cdot]$ denotes the horizontal concatenation operator, $W^O \in \mathbb{R}^{(3 \times d) \times d}$ is a learnable matrix. For $\mathrm{Attention}(\cdot)$, we adopt the attention mechanism introduced by Zhou et al. (2026), which boosts the perception of diverse geometric patterns through an adaptation function $f(\alpha, N, d_{ij})$ with bias weight $\alpha$ (see Appendix F for implementation details). Note that since the relation $r_{ij}$ is scale-independent, we remove the scale $N$ in the adaptation function for $\boldsymbol{R}$. If $\boldsymbol{R}$ is absent, we substitute a zero vector for $\bar{\mathbf{h}}_i^{(2)}$. For a comparison between our MBM and existing attentions (Vaswani et al., 2017; Kwon et al., 2021), please see Appendix G.

**Problem-Conditioned Parameter Generation** Using a single static set of parameters to solve 110 VRP variants is a significant challenge. To mitigate this, we introduce a problem-conditioned mechanism based on problem representation $\boldsymbol{\lambda}$, which consists of two components: (1) $\mathrm{BIAS}(\boldsymbol{\lambda})$ adjusts the degree to which different priors are integrated into the attention; and (2) $\mathrm{WEIGHT}(\boldsymbol{\lambda})$ generates decoder parameters for each problem.

As shown in Figure 3, unlike existing adapter-based fine-tuning methods (Lin et al., 2024; Drakulic et al., 2025), we provide a unified decoding entrance and enable URS to adaptively generate decoder parameters for each problem. This conditional weight generation removes problem-specific adapters, enabling zero-shot generalization to unseen problems while maintaining solution quality.

For MBM in Equation (5), we generate $\alpha$ via a lightweight bias network $\mathrm{BIAS}(\boldsymbol{\lambda})$:

$$\mathrm{BIAS}(\boldsymbol{\lambda}) = \max\left(1, \left(\boldsymbol{\lambda} W_1 + \mathbf{b}_1\right) W_2 + \mathbf{b}_2\right), \quad (6)$$

where $W_1 \in \mathbb{R}^{|\boldsymbol{\lambda}| \times d}$, $W_2 \in \mathbb{R}^{d \times 1}$, $\mathbf{b}_1 \in \mathbb{R}^d$, $\mathbf{b}_2 \in \mathbb{R}^1$ are learnable parameters. We set the minimum bias to 1, resulting in faster model convergence.

For decoder $\boldsymbol{\theta}_{dec}(\boldsymbol{\lambda})$, all parameters are generated based on $\boldsymbol{\lambda}$. Following Kwon et al. (2020), given the first and last

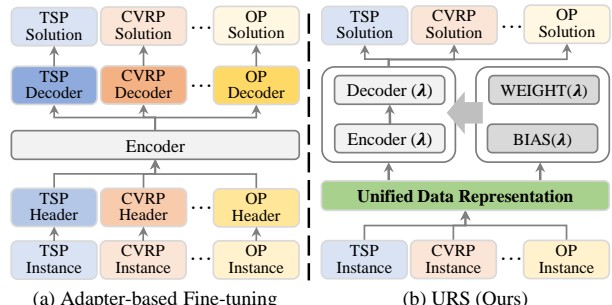

(a) Adapter-based Fine-tuning     (b) URS (Ours)

*Figure 3.* Comparison between our URS and existing adapter-based fine-tuning frameworks.

visited node embeddings $\mathbf{h}_{\pi_1}^{(L)}$ and $\mathbf{h}_{\pi_{t-1}}^{(L)}$, and an optional constraint state $C_t \in \mathbb{R}^1$ (e.g., remaining load), the context embedding $\mathbf{h}_{(C)}^t$ can be expressed as:

$$\mathbf{h}_{(C)}^t = \left[\mathbf{h}_{\pi_1}^{(L)}, \, \mathbf{h}_{\pi_{t-1}}^{(L)}, \, C_t\right] W_Q(\boldsymbol{\lambda}), \quad (7)$$

where $[\cdot, \cdot]$ is the horizontal concatenation operator, $W_Q(\boldsymbol{\lambda}) \in \mathbb{R}^{(2d+1) \times d}$ is a linear projection matrix conditioned on $\boldsymbol{\lambda}$. Missing $C_t$ is zero-padded (see Appendix E.3 for details). The new context embedding $\hat{\mathbf{h}}_{(C)}^t$ is obtained via $\mathrm{Attention}(\cdot)$ (Zhou et al., 2026) on $\mathbf{h}_{(C)}^t$ and $H^{(L)}$:

$$K = H^{(L)} W_K(\boldsymbol{\lambda}), \qquad V = H^{(L)} W_V(\boldsymbol{\lambda}), \quad (8)$$

$$\hat{\mathbf{h}}_{(C)}^t = \mathrm{Attention}(\mathbf{h}_{(C)}^t, K, V, \mathcal{M}_t, f(\alpha, N, d_{t,i})), \quad (9)$$

where $W_K(\boldsymbol{\lambda}), W_V(\boldsymbol{\lambda}) \in \mathbb{R}^{d \times d}$ are linear projection matrices conditioned on $\boldsymbol{\lambda}$. Finally, we compute probabilities for feasible nodes $p_{\boldsymbol{\theta}}(\pi_t = i \mid \mathcal{G}, \pi_{1:t-1}) = \mathrm{softmax}(\mathbf{u})$ based on the improved compatibility (Zhou et al., 2026):

$$u_i^t = \begin{cases} \zeta \cdot \tanh(\frac{\hat{\mathbf{h}}_{(C)}^t (\mathbf{h}_i^{(L)})^{\mathrm{T}}}{\sqrt{d}} + f(\alpha, N, d_{t,i})) & \text{if } i \notin \{\pi_{1:t-1}\} \\ -\infty & \text{otherwise,} \end{cases} \quad (10)$$

where $\zeta$ is the clipping parameter. Inspired by Lin et al. (2022), we use a simple MLP hypernetwork $\mathrm{WEIGHT}(\boldsymbol{\lambda})$ to generate $\{W_Q(\boldsymbol{\lambda}), W_K(\boldsymbol{\lambda}), W_V(\boldsymbol{\lambda})\}$ (see Appendix E.2 for implementation details). For $\alpha$ in Equation (9) and Equation (10), we generate them via $\mathrm{BIAS}(\boldsymbol{\lambda})$ in Equation (6).

## 4. Experiments

In this section, we present a comprehensive evaluation of URS against both classical and neural solvers. For URS, we focus on four key aspects: (1) performance on 11 seen problems; (2) zero-shot generalization on 99 unseen problems; (3) scalability to large-scale VRP variants; and (4) results on benchmark datasets. All experiments are conducted using a single NVIDIA GeForce RTX 4090 GPU (24GB of memory) for neural solvers and an Intel Xeon Gold 6240R CPU (2.40 GHz) for classical solvers.

*Table 2.* Performance comparison on 8 seen less-explored variants.

| Method | OP100 Gap | OP100 Time | PCTSP100 Gap | PCTSP100 Time | PDTSP100 Gap | PDTSP100 Time | ACVRP100 Gap | ACVRP100 Time | CVRPTW100 Gap | CVRPTW100 Time | CVRPB100 Gap | CVRPB100 Time | OCVRP100 Gap | OCVRP100 Time | OCVRPTW100 Gap | OCVRPTW100 Time |
|---|---|---|---|---|---|---|---|---|---|---|---|---|---|---|---|---|
| Oracle | 0.00% | 1.5m | 0.00% | 1.2h | 0.00% | 9.8m | 0.00% | 6.6m | 0.00% | 19.6m | 0.00% | 20.8m | 0.00% | 5.3m | 0.00% | 20.8m |
| Sym-NCO | 0.68% | 15s | 0.14% | 16s | – | | – | | – | | – | | – | | – | |
| BQ | 0.63% | 7s | – | | – | | – | | – | | – | | – | | – | |
| Heter-AM | – | | – | | 6.61% | 5m | – | | – | | – | | – | | – | |
| MVMoE | – | | – | | – | | – | | 4.90% | 15s | 1.28% | 11s | 3.14% | 13s | 3.85% | 15s |
| MTPOMO | – | | – | | – | | – | | 5.31% | 8s | 1.67% | 5s | 3.46% | 6s | 4.41% | 7s |
| ReLD-MTL | – | | – | | – | | – | | 4.56% | 10s | 0.90% | 6s | 2.32% | 7s | 3.10% | 9s |
| GOAL-MTL | 1.20% | 38s | – | | – | | – | | 4.66% | 42s | – | | – | | – | |
| URS-STL | 0.11% | 4s | -0.09% | 5s | 3.22% | 4s | 2.01% | 1.4m | 4.50% | 8s | 1.59% | 6s | 2.25% | 6s | 3.22% | 8s |
| URS | 0.45% | 4s | 1.06% | 5s | 4.98% | 4s | 3.06% | 1.4m | 6.13% | 8s | 1.46% | 6s | 3.24% | 6s | 5.07% | 8s |

*Table 3.* Experimental results on seen ATSP, TSP, and CVRP.

| Method | ATSP100 Gap | ATSP100 Time | TSP100 Gap | TSP100 Time | CVRP100 Gap | CVRP100 Time |
|---|---|---|---|---|---|---|
| Oracle | 0.00% | 2m | 0.00% | 6m | 0.00% | 9.1m |
| Sym-NCO | – | | 0.14% | 5s | 1.46% | 6s |
| BQ | 1.27% | 1m | 0.35% | 7s | 3.24% | 8s |
| LEHD | – | | 0.59% | 2s | 3.68% | 2s |
| POMO | – | | 0.13% | 5s | 1.25% | 6s |
| ICAM | 3.05% | 1m | 0.15% | 4s | 2.04% | 4s |
| MatNet | 0.95% | 6.5m | – | | – | |
| MVMoE | – | | – | | 1.65% | 12s |
| MTPOMO | – | | – | | 1.85% | 6s |
| ReLD-MTL | – | | – | | 1.42% | 8s |
| GOAL-MTL | 1.77% | 1m | – | | 4.22% | 48s |
| URS-STL | 0.62% | 1.1m | 0.08% | 6s | 1.63% | 6s |
| URS | 2.26% | 1.1m | 0.57% | 6s | 1.81% | 6s |

## 4.1. Experimental Setup

**Problem Setting** We train URS on 11 mixed variants to ensure that most features in UDR have been seen at least twice, which prevents URS from overfitting a specific feature to a single problem. Training problems and related data generations are as follows: (1) ATSP (Kwon et al., 2021); (2) TSP (Kool et al., 2019); (3) CVRP (Kool et al., 2019); (4) ACVRP (Kwon et al., 2021; Kool et al., 2019); (5) Orienteering Problem (OP) (Kool et al., 2019); (6) PCTSP (Kool et al., 2019); (7) PDTSP (Li et al., 2021); (8) CVRPTW (Zhou et al., 2024); (9) OCVRP (Zhou et al., 2024); (10) CVRPB (Zhou et al., 2024), and (11) OCVRPTW (Zhou et al., 2024). We provide further discussion of problem selection in Appendix J.1. Each of 110 variants comprises 1,000 instances. The related oracle solvers and problem representations are detailed in Appendix D.

**Model & Training Setting** We use an embedding dimension $d$ of 128 and a feed-forward layer dimension of 512. We set the number of attention layers in the encoder to 12. The clipping parameter $\zeta$ is set to 50 in Equation (10). URS is trained by the REINFORCE (Williams, 1992) algorithm, and the data is generated on the fly (see Appendix E.4 for

more details). We train URS for 500 epochs with 2,000 batches per epoch, and each batch has a size of 128. At each training step, we randomly select one problem from 11 problems. We employ the AdamW optimizer (Loshchilov & Hutter, 2017) with an initial learning rate of $10^{-4}$, which is decayed by a factor of 0.1 at the 451st epoch. Weight decay is set to $10^{-6}$. More details are provided in Appendix H.

**Baseline** We compare URS with the following methods: (1) **Classical Solvers:** PyVRP (Wouda et al., 2024), LKH3 (Helsgaun, 2017), and OR-Tools (Perron & Furnon, 2023). We run LKH3 (Helsgaun, 2017) with 10000 trials and 1 run (Kool et al., 2019). For PyVRP and OR-Tools, we run them on a single CPU core with time limits of 20s (Li et al., 2025a); (2) **Neural Solvers:** (i) *specialist solvers*: Sym-NCO (Kim et al., 2022), BQ (Drakulic et al., 2023), LEHD (Luo et al., 2023), ICAM (Zhou et al., 2026), POMO (Kwon et al., 2020), ELG (Gao et al., 2024), Mat-Net (Kwon et al., 2021), and Heter-AM (Li et al., 2021); (ii) *generalist neural solvers*: GOAL-MTL (Drakulic et al., 2025), MTPOMO (Liu et al., 2024)[1], MVMoE/4E (abbreviated as MVMoE) (Zhou et al., 2024), ReLD-MTL (Huang et al., 2025), RouteFinder (abbreviated as RF) (Berto et al., 2025), and CaDA (Li et al., 2025a)[2].

**Metrics and Inference** We report the optimality gap (Gap) and total inference time (Time) for each method. The gap quantifies the discrepancy between the corresponding solvers and the oracle solver. For neural solvers, we execute the source code provided by the authors using default settings. We compare our URS with relevant baselines for each problem, as well as its single-task version (i.e., URS-STL). The URS-STL variant uses exactly the same architecture and hyperparameters (see Appendix H.2 for more details). For URS, we report the best result with ×8 (Kwon et al., 2020) and ×128 (Kwon et al., 2021) instance augmentation

---

[1] For MTPOMO, we use the implementation in Zhou et al. (2024), ensuring problem consistency across all neural solvers.

[2] Given the disparities in problem and training settings, detailed comparisons against RF and CaDA are deferred to Appendix J.2.

*Table 4.* Zero-shot generalization on 1K test instances of 32 unseen VRPs ($N = 100$). Due to page limit, complete experimental results (including all 99 unseen variants) are deferred to Appendix I.

| Method | CVRPBP | | CVRPBPL | | MDOCVRPBP | | MDOCVRPBPTW | | MDOCVRPBPL | | MDOCVRPBPLTW | | MDCVRP | | MDOCVRP | |
|---|---|---|---|---|---|---|---|---|---|---|---|---|---|---|---|---|
| | Gap | Time | Gap | Time | Gap | Time | Gap | Time | Gap | Time | Gap | Time | Gap | Time | Gap | Time |
| Oracle | 0.00% | 6.6m | 0.00% | 6.6m | 0.00% | 6.6m | 0.00% | 6.6m | 0.00% | 6.6m | 0.00% | 6.6m | 0.00% | 6.6m | 0.00% | 6.6m |
| MVMoE | 13.95% | 14s | 13.38% | 14s | 56.60% | 33s | 63.77% | 38s | 56.26% | 34s | 62.47% | 40s | 33.06% | 31s | 29.08% | 31s |
| MTPOMO | 13.71% | 7s | 13.44% | 8s | 42.94% | 24s | 47.09% | 29s | 41.98% | 27s | 46.26% | 32s | 23.38% | 23s | 27.46% | 23s |
| ReLD-MTL | 13.57% | 9s | 12.96% | 10s | 37.74% | 27s | 45.16% | 34s | 37.45% | 29s | 44.74% | 36s | 18.39% | 25s | 22.32% | 25s |
| URS | **12.95%** | 7s | **12.65%** | 8s | **24.44%** | 22s | **26.31%** | 26s | **24.22%** | 24s | **26.20%** | 28s | **15.05%** | 23s | **22.01%** | 23s |

| Method | MDCVRPL | | MDOCVRPTW | | MDOCVRPB | | MDOCVRPBL | | MDOCVRPLTW | | SPCTSP | | PDCVRP | | OPDCVRP | |
|---|---|---|---|---|---|---|---|---|---|---|---|---|---|---|---|---|
| | Gap | Time | Gap | Time | Gap | Time | Gap | Time | Gap | Time | Gap | Time | Gap | Time | Gap | Time |
| Oracle | 0.00% | 6.6m | 0.00% | 6.6m | 0.00% | 6.6m | 0.00% | 6.6m | 0.00% | 6.6m | 0.00% | 1.6h | 0.00% | 1.6h | 0.00% | 1.6h |
| MVMoE | 33.42% | 34s | 51.57% | 40s | 28.21% | 26s | 28.70% | 28s | 51.09% | 43s | – | | – | | – | |
| MTPOMO | 24.32% | 25s | 35.71% | 29s | 25.06% | 20s | 25.01% | 21s | 35.39% | 31s | – | | – | | – | |
| ReLD-MTL | 18.48% | 28s | 33.55% | 33s | 18.50% | 21s | 18.50% | 22s | 33.26% | 35s | – | | – | | – | |
| URS | **15.32%** | 26s | **26.05%** | 25s | **15.17%** | 18s | **14.90%** | 20s | **25.59%** | 27s | **-2.37%** | 5s | **-1.47%** | 5s | **4.93%** | 5s |

| Method | ACVRPL | | ACVRPBL | | ACVRPB | | ACVRPBTW | | APDTSP | | AMDCVRPL | | AMDCVRP | | APDCVRP | |
|---|---|---|---|---|---|---|---|---|---|---|---|---|---|---|---|---|
| | Gap | Time | Gap | Time | Gap | Time | Gap | Time | Gap | Time | Gap | Time | Gap | Time | Gap | Time |
| Oracle | 0.00% | 6.6m | 0.00% | 6.6m | 0.00% | 6.6m | 0.00% | 6.6m | 0.00% | 6.6m | 0.00% | 6.6m | 0.00% | 6.6m | 0.00% | 6.6m |
| URS | **-2.24%** | 1.9m | **6.18%** | 2.4m | **7.10%** | 2.1m | **9.95%** | 2.2m | **6.21%** | 1m | **7.15%** | 6.1m | **8.11%** | 5.3m | **7.03%** | 1.2m |

| Method | ACVRPLTW | | ACVRPBLTW | | ACVRPBPLTW | | AOCVRPBLTW | | ACVRPTW | | ACVRPBPTW | | AOPDCVRP | | AOP | |
|---|---|---|---|---|---|---|---|---|---|---|---|---|---|---|---|---|
| | Gap | Time | Gap | Time | Gap | Time | Gap | Time | Gap | Time | Gap | Time | Gap | Time | Gap | Time |
| Oracle | 0.00% | 6.6m | 0.00% | 6.6m | 0.00% | 6.6m | 0.00% | 6.6m | 0.00% | 6.6m | 0.00% | 6.6m | 0.00% | 6.6m | 0.00% | 6.6m |
| URS | **12.22%** | 2.7m | **10.18%** | 2.4m | **13.82%** | 2.8m | **14.32%** | 2.5m | **11.28%** | 2.5m | **14.52%** | 2.6m | **11.13%** | 1.2m | **-5.47%** | 1.2m |

*Table 5.* Generalization on different large-scale VRP variants. All models are trained on instances of size $N = 100$ and employ instance augmentation $\times 8$. PyVRP and OR-Tools can process all instances of a given size concurrently to ensure a fair comparison.

| Problem | Method | N = 1000 | | N = 2000 | | N = 3000 | | N = 4000 | | N = 5000 | |
|---|---|---|---|---|---|---|---|---|---|---|---|
| | | Obj.(Gap) | Time | Obj.(Gap) | Time | Obj.(Gap) | Time | Obj.(Gap) | Time | Obj.(Gap) | Time |
| CVRPTW | PyVRP | 159.35(0.00%) | 20m | 337.15(0.00%) | 40m | 516.97(0.00%) | 1h | 618.58(0.00%) | 2.7h | 787.25(0.00%) | 3.3h |
| | OR-Tools | 178.13(11.79%) | 20m | 357.18(5.94%) | 40m | 533.97(3.29%) | 1h | 647.92(4.74%) | 2.7h | 807.128(2.52%) | 3.3h |
| | MTPOMO | 219.22(37.57%) | 1.7m | 488.40(44.86%) | 14.6m | 773.00(49.53%) | 49.5m | 975.61(57.72%) | 1.9h | OOM | |
| | MVMoE | 233.53(46.55%) | 1.7m | 624.73(85.30%) | 14.6m | 1094.25(111.67%) | 51.9m | 1486.34(140.28%) | 2h | OOM | |
| | ReLD-MTL | 183.31(15.04%) | 1.6m | 396.28(17.54%) | 13.4m | 613.58(18.69%) | 45.7m | 748.65(21.03%) | 1.8h | OOM | |
| | URS | **172.58(8.30%)** | **1.3m** | **365.27(8.34%)** | **11.1m** | **559.03(8.14%)** | **30.1m** | **673.63(8.90%)** | **1.2h** | **853.79(8.45%)** | **2.3h** |
| CVRPB | PyVRP | 36.52(0.00%) | 20m | 51.80(0.00%) | 40m | 75.12(0.00%) | 1h | 93.52(0.00%) | 2.7h | 112.21(0.00%) | 3.3h |
| | OR-Tools | 44.23(21.11%) | 20m | 63.33(22.26%) | 40m | 88.6(17.94%) | 1h | 108.82(16.36%) | 2.7h | 128.11(14.17%) | 3.3h |
| | MTPOMO | 45.92(25.77%) | 1.1m | 83.50(61.20%) | 8.8m | 136.52(81.73%) | 31.4m | 197.99(111.70%) | 1.2h | OOM | |
| | MVMoE | 98.60(170.03%) | 1m | 250.47(383.57%) | 8.1m | 332.93(343.19%) | 28.9m | 423.19(352.49%) | 1.1h | OOM | |
| | ReLD-MTL | 38.52(5.48%) | 1m | 59.21(14.30%) | 8m | 88.36(17.62%) | 28.3m | 114.88(22.83%) | 1.1h | OOM | |
| | URS | **34.86(-4.54%)** | **40s** | **50.39(-2.72%)** | **5.1m** | **72.01(-4.15%)** | **17.3m** | **91.74(-1.91%)** | **41.2m** | **111.93(-0.24%)** | **1.3h** |
| OCVRPTW | PyVRP | 90.91(0.00%) | 20m | 164.00(0.00%) | 40m | 224.16(0.00%) | 1h | 299.45(0.00%) | 2.7h | 367.86(0.00%) | 3.3h |
| | OR-Tools | 105.98(16.58%) | 20m | 207.041(26.24%) | 40m | 280.88(25.30%) | 1h | 370.85(23.84%) | 2.7h | 447.02(21.52%) | 3.3h |
| | MTPOMO | 147.14(61.85%) | 1.6m | 297.53(81.43%) | 12.8m | 437.73(95.28%) | 44.3m | 601.63(100.91%) | 1.7h | OOM | |
| | MVMoE | 147.35(62.09%) | 1.6m | 389.60(137.57%) | 12.9m | 634.11(182.88%) | 44.7m | 899.18(200.28%) | 1.7h | OOM | |
| | ReLD-MTL | 110.00(21.01%) | 1.5m | 208.60(27.20%) | 12.3m | 292.70(30.57%) | 42m | 394.42(31.72%) | 1.6h | OOM | |
| | URS | **98.76(8.63%)** | **1m** | **178.07(8.58%)** | **7.6m** | **243.88(8.80%)** | **25.4m** | **325.64(8.74%)** | **1h** | **398.72(8.39%)** | **1.9h** |

for symmetric and asymmetric instances, respectively. The mark $(-)$ indicates that the method does not support the corresponding VRP variant.

## 4.2. Performance Evaluation

**Performance on Seen VRPs** As presented in Table 2 and Table 3, among comparable neural solvers, URS-STL achieves the lowest gap on 8 of the 11 VRPs while offering fast inference, highlighting the strength of our architecture when fully specialized. Although URS-MTL exhibits a performance degradation relative to STL variants, it remains competitive across all 11 problems. Using ATSP as an exam-

ple, we attribute this drop to geometric structural inequality, which likely biases the model's learning capacity toward symmetric VRPs. While the lack of explicit constraint states yields weaker performance compared to ReLD-MTL and MVMoE, URS shows clear advantages when addressing unseen VRPs (Table 4) and large-scale instances (Table 5).

**Generalization to Unseen VRPs** We evaluate our URS on zero-shot generalization across 99 unseen VRP variants. As presented in Table 4 and Appendix I, on 44 widely-studied VRPs with diverse constraint combinations (i.e., C, O, L, TW, MD, B, BP), URS continues to deliver high-

*Table 6.* Comparison on Set-AGS with $N \in [3000, 7000]$. All models are trained on instances of size $N = 100$.

| Type | Method | Leuven1 (N=3000) | Leuven2 (N=4000) | Antwerp1 (N=6000) | Antwerp2 (N=7000) | Avg.gap |
|------|--------|--------|--------|--------|--------|---------|
| Specialist | LEHD greedy | 16.60% | 34.86% | 14.66% | 22.77% | 22.22% |
| | BQ greedy | 18.53% | 30.70% | 16.48% | 27.67% | 23.34% |
| | POMO aug×8 | 460.32% | 202.17% | OOM | OOM | – |
| | ELG aug×8 | 12.12% | 21.52% | OOM | OOM | – |
| Generalist | MTPOMO aug×8 | 67.72% | 87.31% | OOM | OOM | – |
| | MVMoE/4E aug×8 | 299.10% | 170.10% | OOM | OOM | – |
| | MVMoE/4E-L aug×8 | 182.28% | 127.07% | OOM | OOM | – |
| | RF-MVMoE aug×8 | 57.30% | 160.27% | OOM | OOM | – |
| | RF-TE aug×8 | 26.90% | 45.56% | OOM | OOM | – |
| | CaDA aug×8 | 1035.51% | OOM | OOM | OOM | – |
| | ReLD-MTL aug×8 | 17.00% | 30.59% | OOM | OOM | – |
| | GOAL greedy | OOM | OOM | OOM | OOM | – |
| | URS aug×8 | **11.50%** | **17.61%** | **9.18%** | **14.86%** | **13.29%** |

quality solutions. Notably, URS merely selects the next node from a given set of feasible candidates without any additional domain knowledge, making its performance especially meaningful. Benefiting from our UDR, URS can directly address 55 less-explored VRP variants that compound asymmetry and PD relations. These results highlight URS's strong capacity for cross-problem zero-shot generalization without any fine-tuning.

**Scalability to Larger-scale VRP Variants**   We conduct experiments on large-scale instances to validate URS's cross-problem scalability. Specifically, we generate five large-scale $(1{,}000 - 5{,}000$ nodes) synthetic datasets for each variant (CVRPTW, CVRPB, and OCVRPTW), and each dataset comprises 16 instances. We adhere to the capacity constraints specified in TAM (Hou et al., 2022). As shown in Table 5, URS consistently achieves the best solution quality, complemented by remarkably fast inference speeds, across various problem instances among all comparable multi-task neural solvers. Despite a significant increase in the problem scale (up to 50× the training size, $N = 100$), the performance of URS remains stable, with no severe degradation. Remarkably, URS already outperforms the strong PyVRP with a carefully manual design on large-scale CVRPB instances. These extensive results show URS's strong scalability to large-scale complex VRP variants.

**Results on Benchmark Dataset**   We further evaluate URS on benchmark instances from CVRPLib Set-X (Uchoa et al., 2017) and CVRPLib Set-AGS (also known as Set-XXL) (Arnold et al., 2019). We compare URS against representative specialist and generalist solvers. As presented in Table 6, URS can significantly outperform existing generalist neural solvers in large-scale benchmark instances. Notably, as a generalist solver, URS also greatly surpasses many representative specialist models (e.g., LEHD, BQ, and ELG). In addition, the detailed results on the Set-X dataset (Appendix J.3) show that URS maintains its position as the best-performing overall. These results further underscore the applicability of URS in real-world large-scale scenarios.

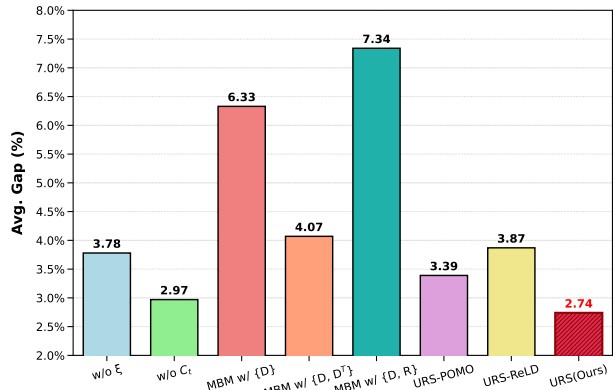

*Figure 4.* Effects of different components of URS.

*Table 7.* Results comparing performance with and without MBM, WEIGHT($\boldsymbol{\lambda}$), and BIAS($\boldsymbol{\lambda}$) on seen and unseen variants.

| MBM | WEIGHT($\boldsymbol{\lambda}$) | BIAS($\boldsymbol{\lambda}$) | Avg.gap Seen (11) | Avg.gap Unseen (11) | Best Solution |
|-----|-----|-----|-----|-----|-----|
| × | ✓ | ✓ | 6.33% | 10.41% | 3/22 |
| ✓ | × | × | 4.20% | 8.12% | 0/22 |
| ✓ | × | ✓ | 3.84% | 8.02% | 1/22 |
| ✓ | ✓ | × | 3.63% | 7.81% | 1/22 |
| ✓ | ✓ | ✓ | **2.74%** | **7.05%** | **17/22** |

## 5. Ablation Study

In this section, we conduct a comprehensive ablation study and analysis to demonstrate the effectiveness and robustness of URS. The detailed results and discussion are presented below. Note that, unless otherwise stated, the instance augmentation technique is used for the ablation studies.

**Effects of Key Components**   We conduct an ablation study across 11 seen VRP variants to validate the effect of key components of URS, mainly including: (1) without node-type indicator $\boldsymbol{\xi}$ of UDR; (2) without problem state $C_t$; (3) effects of different priors in MBM ; (4) effects of alternative decoder architectures. As detailed in Figure 4, URS maintains its competitive performance without $C_t$; its effectiveness is further improved by the inclusion of a node-type indicator $\boldsymbol{\xi}$. For MBM, the results reveal that three priors $\{\boldsymbol{D}, \boldsymbol{D}^{\mathrm{T}}, \boldsymbol{R}\}$ are complementary, and their combined use allows MBM to effectively learn the geometric and relational biases inherent in various problems, resulting in superior overall performance. Furthermore, replacing the decoder with POMO or ReLD confirms the strong cross-problem robustness of URS, while integrating the decoder introduced in Zhou et al. (2026) further improves cross-problem performance. More detailed results and analyses can be found in Appendix K.

**MBM & Parameter Generator**   To further evaluate the effects of MBM, WEIGHT($\boldsymbol{\lambda}$), and BIAS($\boldsymbol{\lambda}$) for cross-

*Table 8.* Performance comparison of ablating each UDR component (i.e., $\rho$, $\omega$, and $\xi$) on seen and unseen variants. Since removing positional identifiers $\rho$ prevents the use of instance augmentation, this augmentation technique is disabled across all results to maintain a consistent inference strategy.

| $\rho$ | $\omega$ | $\xi$ | Avg.gap Seen (11) | Avg.gap Unseen (11) | Best Solution |
|---|---|---|---|---|---|
| $\times$ | $\checkmark$ | $\checkmark$ | 4.65% | 18.12% | 3/22 |
| $\checkmark$ | $\times$ | $\checkmark$ | 9.81% | 18.60% | 0/22 |
| $\checkmark$ | $\checkmark$ | $\times$ | 5.43% | 18.60% | 4/22 |
| $\checkmark$ | $\checkmark$ | $\checkmark$ | **4.17%** | **8.51%** | **15/22** |

problem zero-shot generalization, we conduct tests on 11 seen problems and 11 unseen CVRP variants widely studied in (Liu et al., 2024). As shown in Table 7, the full URS model achieves the lowest average gap on both seen and unseen problems, and provides the best solution on the majority of tasks (17/22). This shows that their synergy is essential for robust cross-problem zero-shot generalization. We provide detailed results in Appendix K.5.

**Effects of Each UDR Component**   To clarify the independent contribution of each UDR component, we conduct ablations by removing each component in turn. As shown in Table 8, the absence of any single component leads to notable performance degradation. A complete UDR achieves the lowest average gap, affirming the necessity of our UDR design for robust zero-shot generalization. The detailed results for each VRP variant can be found in Appendix K.6.

**Hypernetwork vs. Adapter**   We train URS-Adapter, a URS variant that adopts a distinct decoder for each of the 11 training problems, and compare it with the original URS with a hypernetwork on the same 11 seen problems and large-scale CVRP variants (see Appendix K.7 for detailed results). URS consistently outperforms URS-Adapter across the 11 problems seen, achieving higher parameter efficiency. This performance advantage holds consistently across large-scale CVRP variants, further solidifying our conclusion.

## 6. Conclusion, Limitation, and Future Work

In this work, we propose a novel RL-based URS for cross-problem zero-shot generalization. The core of URS is the proposed UDR, which broadens problem coverage by substituting problem enumeration with data unification. Its generalization capability is further enhanced by an MBM and a problem-conditioned parameter generator. Extensive experiments show that URS can produce high-quality solutions for 110 VRP variants without any fine-tuning, including 99 unseen variants, significantly broadening the problem coverage and reducing reliance on domain expertise.

**Limitation and Future Work**   While URS achieves impressive results across a wide range of VRP variants, its current training scheme does not fully account for the inherent disparities in problem complexity. In the future, we aim to explore more effective strategies, such as hierarchical training and difficulty-adaptive sampling, to improve training efficiency. Additionally, extending our method to accommodate more complex variants (e.g., dynamic VRP) is a promising direction for future research.

## Acknowledgements

This work was supported by the National Natural Science Foundation of China (Grant No. 62476118), the Guangdong Provincial Key Laboratory of Fully Actuated System Control Theory and Technology (Grant No. 2024B1212010002), the Natural Science Foundation of Guangdong Province (Grant No. 2024A1515011759), and the Center for Computational Science and Engineering at Southern University of Science and Technology. We would like to thank the anonymous reviewers and (S)ACs of ICML 2026 for their constructive comments and dedicated service to the community. Additionally, we would like to thank Mr. Rongsheng Chen for his valuable advice and assistance during this research.

## Impact Statement

This paper presents work whose goal is to advance the field of Machine Learning. There are many potential societal consequences of our work, none of which we feel must be specifically highlighted here.

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

# A. Related Work

## A.1. Neural Routing Solvers with Single-Task Learning

The NCO methods have emerged as a promising paradigm for tackling various routing problems, achieving impressive empirical performance. Early seminal work based on Pointer Networks ignited interest in learning end-to-end construction policies for TSP and CVRP (Vinyals et al., 2015; Bello et al., 2016; Nazari et al., 2018). Subsequently, the Transformer-style heavy-encoder and light-decoder auto-regressive architecture has become the dominant paradigm (Kool et al., 2019), while POMO (Kwon et al., 2020) exploits permutation symmetry to enhance exploration diversity. Many advancements within this paradigm have pushed promising results on small instances (e.g., 100-node TSPs) (Xin et al., 2020; Kim et al., 2022; Xiao et al., 2024a;b; Sun et al., 2024). However, models trained on a fixed instance scale often degrade sharply when evaluated out of scale. To mitigate the poor generalization performance, recent studies introduce auxiliary local policies (Gao et al., 2024), adaptive perception across instance scales (Zhou et al., 2026), or different search space reduction methods (Fang et al., 2024; Zhou et al., 2025; Chen et al., 2025). In parallel, "heavy decoder" designs that dynamically recompute embeddings at each construction step (Drakulic et al., 2023; Luo et al., 2023) demonstrate strong generalization on large-scale instances. To improve the performance on large-scale routing instances, complementary directions are reducing solving difficulty via problem decomposition (Zheng et al., 2024b; Ye et al., 2024; Li et al., 2025b) or adopting non-autoregressive generation augmented with additional searches (e.g., Monte Carlo tree search (MCTS) and 2-opt) (Fu et al., 2021; Sun & Yang, 2023; Qiu et al., 2022; Li et al., 2023; Ye et al., 2023). Moreover, heterogeneous vehicle routing (Li et al., 2022), asymmetric distance settings (Kwon et al., 2021), multiple VRPs (Zheng et al., 2024a), and pickup-delivery problems (Li et al., 2021; Kong et al., 2024) have each motivated tailored neural designs, which have yielded competitive results. Despite these advancements, they still train a specialist model per problem type, limiting cross-problem transfer. We instead focus on a unified neural routing solver that aims to generalize across multiple routing problems without retraining from scratch for each specification.

## A.2. Neural Routing Solvers with Multi-Task Learning

To address the challenge of cross-problem generalization, growing attention has shifted to multi-task learning capable of adapting to diverse routing problems. Existing efforts can broadly fall into two categories: (1) constraint combination-based multi-task learning and (2) adapter-based fine-tuning frameworks. The former line (e.g., MTPOMO (Liu et al., 2024)) treats VRP variants as combinations of a predefined set of attributes and trains a shared backbone, achieving promising results on up to 16 VRP variants. Follow-up work extends this paradigm by introducing mixture-of-experts (MoE) modules (Zhou et al., 2024), heavy decoder designs plus knowledge distillation (Zheng et al., 2025), and constraint-aware dual-attention (Li et al., 2025a) or updated Transformer structure (Berto et al., 2025) to improve model performance. While this enables knowledge sharing across problems, its coverage is inherently bounded by the hand-specified attribute set. The second family trains a shared backbone and performs adaptation to each variant via lightweight adapters. For instance, Lin et al. (2024) pretrain a TSP backbone and adapt to new variants through adapter-based fine-tuning. Drakulic et al. (2025) integrate edge features and adopt supervised multi-task pretraining for obtaining a powerful backbone before fine-tuning on novel tasks. Wang et al. (2025a) propose a multi-armed bandit strategy to realize more efficient multi-task training. Although these reduce retraining cost, they fail to achieve zero-shot generalization. In contrast, we focus on a constructive neural routing solver supported with zero-shot capability: once trained, it can directly generate high-quality solutions for previously unseen problems without any fine-tuning.

## B. Setups of VRP Variants

In this paper, we consider 110 VRP variants which may simultaneously have one or more constraints from the following categories: (1) Capacity (C); (2) Open Route (O); (3) Backhaul (B); (4) Backhaul and Priority (BP); (5) Duration Limit (L); (6) Time Windows (TW); (7) Multi-Depot (MD); (8) Prize Collecting (PC); (9) Asymmetric (A); (10) Pickup and Delivery (PD). The node coordinates are randomly sampled from the unit square when available (in symmetric geometric cases, i.e., $d_{ij} = d_{ji}$), following AM (Kool et al., 2019). Below, we provide a comprehensive description of the data generation process for each constraint.

**Capacity (C)**   The selection of the vehicle capacity $\mathcal{C}$ is critical, as prior work (Luo et al., 2025c) demonstrates its significant impact on the solution structure. For PDVRP variants (e.g., PDCVRP, OPDCVRP, APDCVRP, and AOPDCVRP), an excessively large $\mathcal{C}$ causes the solution to approximate that of a PDTSP instance. Conversely, an overly restrictive $\mathcal{C}$ promotes a simplistic, repetitive Pickup-Delivery sequence. To ensure structurally sound solutions, we set $\mathcal{C}$=20 for PDCVRP variants. For all other VRP variants, we adopt the convention from AM (Kool et al., 2019), setting $\mathcal{C}$=50.

**Open Route (O)**   The open route setting implies that vehicles are not required to return to the depot after completing their service tours. This does not alter the data generation, and it specifically affects the final cost evaluation by excluding the travel distance from the last customer back to the depot.

**Backhaul (B)**   The standard CVRP involves vehicles delivering goods from a depot to a set of customers with positive demands. The introduction of backhauls extends this formulation to the CVRP with Backhauls (CVRPB), where vehicles must also load goods from backhaul customers and return them to the depot, in addition to serving the standard linehaul (delivery) customers. Following the experimental setup of MTPOMO (Liu et al., 2024), we generate customer demands by sampling from a discrete uniform distribution over the integers $\{1, \ldots, 9\}$. Subsequently, 20% of these nodes are randomly designated as backhaul customers, and their corresponding demand values are negative, while the remainder are classified as linehaul customers. Notably, no precedence constraints are imposed between the servicing of linehaul and backhaul customers. To ensure the feasibility of the solutions constructed by our neural solver, we adhere to the MVMoE (Zhou et al., 2024) framework's policy that the first customer visited must be a linehaul node.

**Backhaul and Priority (BP)**   This constraint imposes a strict precedence requirement on the standard CVRPB, dictating that all linehaul customers must be serviced prior to any backhaul customers. Consequently, once a vehicle serves a backhaul customer, it is prohibited from visiting any subsequent linehaul nodes within the same route. If the first customer served is a backhaul node, the vehicle's load is initialized to zero. This ensures the neural solver can construct a feasible solution.

**Duration Limit (L)**   Following MTPOMO (Liu et al., 2024), we impose a duration limit (i.e., the maximum length of each vehicle route) of 3 in symmetric cases. Since the node coordinates are randomly sampled from the unit square, the maximum possible distance between any two nodes is $\sqrt{2}$. The setting is sufficient to ensure a vehicle can deliver goods to at least one customer and return to the depot, thus guaranteeing the feasibility of the solution. For the setting of asymmetric cases, we provide details in the description of the following asymmetric instances.

**Time Windows (TW)**   Consistent with the MVMoE (Zhou et al., 2024), we set time windows of the depot to $[e_0, l_0]$=[0,3] and its service time to 0. The service time $s_i$ for each customer node is set to 0.2. For the time windows for each customer node, we generate them following the methodology outlined in MVMoE (Zhou et al., 2024).

**Multi-Depot (MD)**   The generation of coordinates for the multiple depots aligns with that of other nodes, with the number of depots set to 3 (Berto et al., 2025). A vehicle is required to return to the same depot it departed from. To guarantee that each starting node is considered in this multi-depot variant, we set the number of trajectories for an instance to be the product of the number of depots and the number of customer nodes, maximizing the exploration of high-quality solutions.

**Prize Collecting (PC)**   Under the Prize Collecting condition, each node (excluding the depot) is assigned both a prize and a penalty. A prize is collected for each visited node, while a penalty is incurred for each unvisited node. The objective is to minimize the total travel distance plus the sum of penalties for all unvisited nodes. A key constraint is that a minimum amount of total prize must be collected before the vehicle is permitted to return to the depot and terminate its route. In this context, the minimum prize requirement for the four examined problems (PCTSP, SPCTSP, APCTSP, and ASPCTSP) is

standardized to 1.0. Consistent with the AM (Kool et al., 2019) framework, the prizes and penalties are generated as follows: $prize \sim \text{Uniform}\left(0, \frac{4}{n}\right)$, and $penalty \sim \text{Uniform}\left(0, 3 \cdot \frac{k_n}{n}\right)$, where $n = 100$ and $k_n = 4$ in our experiments. For the OP, the objective is to maximize collected prizes without exceeding a predefined distance limit (set to 4.0 for OP and 1.0 for AOP), where unvisited nodes carry no penalties. The prize for each node is assigned proportional to its Euclidean distance from the depot. By normalizing distances to the instance's maximum distance and discretizing them into $[0.01, 1.00]$, nodes further from the depot systematically yield higher rewards, with the depot's prize set to zero.

**Asymmetric (A)**    We adopt the data generation method from MatNet (Kwon et al., 2021) to create an asymmetric distance matrix. Due to the small pairwise distances in the generated distance matrix, the parameters for duration limit and time windows set for symmetric cases are not applicable to the asymmetric problems. To ensure the constraints remain valid, we adjust the duration limit and depot end time to 0.6 and 1.0, respectively.

**Pickup and Delivery (PD)**    For PDP variants, we adhere to the methodology in Li et al. (2021). For a problem of size $N$, node 0 is set to the depot, nodes 1 to $N/2$ are designated as pickup nodes, while nodes from $N/2 + 1$ to $N$ are designated as delivery nodes. For each pickup node $v_i$, its corresponding delivery node is $v_{i+N/2}$. A strict precedence constraint is imposed in PDP cases, requiring each vehicle to visit a pickup node before its corresponding delivery node. For PDCVRP variants, the demand for each delivery node is randomly sampled from a discrete uniform distribution over $\{1, 2, \ldots, 9\}$. The demand for each pickup node is then set to the negative of its corresponding delivery node's demand. To ensure the feasibility of the constraints, the vehicle capacity is set to 20.

## C. Unified Data Representation

In this section, we provide a detailed description of our UDR $U = \{\mathbf{u}_i\}_{i=0}^n$ (see Table 9). Each node $\mathbf{u}_i$ consists of three components, i.e., $\mathbf{u}_i = \{\boldsymbol{\rho}_i,\ \boldsymbol{\omega}_i,\ \boldsymbol{\xi}_i\}$, where $\boldsymbol{\rho}_i,\ \boldsymbol{\omega}_i,\ \boldsymbol{\xi}_i$ are a positional identifier, unified attribute set, and node-type indicator, respectively. To distinguish linehaul/delivery from backhaul/pickup nodes, we assign negative demand values to backhaul/pickup nodes (positive for linehaul/delivery), making demand the only signed component, following (Liu et al., 2024; Zhou et al., 2024).

*Table 9.* Detailed description of each component in UDR.

| Attribute Name | Symbol | Range | Description | Examples |
|---|---|---|---|---|
| Random Identifier | $\eta$ | $[0, 1]$ | Random scalar for asymmetric problems. | ATSP |
| Node Coordinates | $\{x, y\}$ | $[0, 1]^2$ | 2D coordinates of the node. | TSP, CVRP |
| Demand | $\delta$ | $[-1, 1]$ | Node demand ($+$ for delivery, $-$ for pickup). | CVRP, PDCVRP |
| Prize | $\epsilon$ | $[0, 1]$ | Prize collected for each node. | OP, PCTSP |
| Penalty | $\mu$ | $[0, 1]$ | Penalty incurred for each node. | PCTSP |
| Earliest Arrival Time | $e$ | $[0, 1]$ | Earliest permissible arrival time. | CVRPTW |
| Latest Arrival Time | $l$ | $[0, 1]$ | Latest permissible arrival time (deadline). | CVRPTW |
| Service Time | $s$ | $[0, 1]$ | Time required for service at the node. | CVRPTW |
| Depot | $-$ | $\{0, 1\}$ | Indicates if the node is a depot. | CVRP |
| Pickup Node | $-$ | $\{0, 1\}$ | Indicates if the node is a pickup location. | PDTSP |
| Delivery Node | $-$ | $\{0, 1\}$ | Indicates if the node is a delivery location. | CVRP, PDTSP |
| Node in Sub-routes | $-$ | $\{0, 1\}$ | The solution $\pi$ can have sub-routes. | CVRP |
| Node in Open Route | $-$ | $\{0, 1\}$ | Vehicles need not return to the depot. | OCVRP |

# D. Multi-hot Problem Representation

Leveraging the UDR, each problem instance activates only a subset of the unified feature space. Thus, for any instance $\mathcal{G}_i$ from different problems, we simply mark active (non-zero) feature slots with 1 to obtain a corresponding multi-hot problem representation $\boldsymbol{\lambda}_i$. Different from raw feature values, it encodes presence only and provides conditional signals to the following position bias weight (see Equation (6)) and adaptive decoder (see Equation (9) and Equation (10)), helping the model distinguish variants and refine node selection. Here, we provide detailed representations for each problem in our experiments, and each problem is a 13-dimensional multi-hot vector.

As shown in Table 10, we train URS on eleven classic and varied routing problems to ensure that each feature in the UDR has been seen at least once, i.e., $\mathcal{P} = \{P_i\}_{i=1}^{11}$. We adhere to the principle of minimal redundancy and strictly align with existing work on CVRP variant selection (e.g., MVMoE (Zhou et al., 2024) and ReLD (Huang et al., 2025)): most features in UDR appear in at least two training problems, helping prevent URS from overfitting to a single problem. The only exception is the Penalty attribute, as there is structural similarity between the seen PCTSP and unseen SPCTSP. For more detailed analysis and experimental results, please see Appendix J.1.

Then, we evaluate generalization on unseen problems on different combinations of ten constraint attributes (see Appendix B), and the detailed representations of each unseen problem can be found in Table 11 (unseen well-studied VRP variants) and Table 12 (unseen less-explored VRP variants).

*Table 10.* The representations for seen routing problems in our experiments. The total number of problems is 11. In training, each feature in the UDR has been seen at least once. Here, RI represents the random identifier, and EAT, LAT, and ST indicate the earliest arrival time, the latest arrival time, and the service time, respectively.

| Problem | Oracle | RI | Coord. | Demand | Prize | Penalty | EAT | LAT | ST | Depot | Pickup | Delivery | Sub-routes | Open Route |
|---------|--------|-----|--------|--------|-------|---------|-----|-----|-----|-------|--------|----------|-----------|-----------|
| ATSP | LKH-3 | ✓ | | | | | | | | | | | | |
| TSP | LKH-3 | | ✓ | | | | | | | | | | | |
| OP | Compass | | ✓ | | ✓ | | | | | ✓ | | | | |
| PCTSP | ILS | | ✓ | | ✓ | ✓ | | | | ✓ | | | | |
| PDTSP | LKH-3 | | ✓ | | | | | | | ✓ | ✓ | ✓ | | |
| ACVRP | PyVRP | ✓ | | ✓ | | | | | | ✓ | | ✓ | ✓ | |
| CVRP | PyVRP | | ✓ | ✓ | | | | | | ✓ | | ✓ | ✓ | |
| CVRPTW | PyVRP | | ✓ | ✓ | | | ✓ | ✓ | ✓ | ✓ | | ✓ | ✓ | |
| CVRPB | OR-Tools | | ✓ | ✓ | | | | | | ✓ | ✓ | ✓ | ✓ | |
| OCVRP | LKH-3 | | ✓ | ✓ | | | | | | ✓ | | ✓ | ✓ | ✓ |
| OCVRPTW | OR-Tools | | ✓ | ✓ | | | ✓ | ✓ | ✓ | ✓ | | ✓ | ✓ | ✓ |
| Frequency | | 2/11 | 9/11 | 6/11 | 2/11 | 1/11 | 2/11 | 2/11 | 2/11 | 9/11 | 2/11 | 7/11 | 6/11 | 2/11 |

*Table 11.* The problem representations of unseen 44 well-studied VRP variants. Here, RI represents the random identifier, and EAT, LAT, and ST indicate the earliest arrival time, the latest arrival time, and the service time, respectively.

| Problem | Oracle | RI | Coord. | Demand | Prize | Penalty | EAT | LAT | ST | Depot | Pickup | Delivery | Sub-routes | Open Route |
|---|---|---|---|---|---|---|---|---|---|---|---|---|---|---|
| CVRPL | LKH-3 | | ✓ | ✓ | | | | | | ✓ | | ✓ | ✓ | |
| OCVRPB | OR-Tools | | ✓ | ✓ | | | | | | ✓ | ✓ | ✓ | ✓ | ✓ |
| CVRPBL | OR-Tools | | ✓ | ✓ | | | | | | ✓ | ✓ | ✓ | ✓ | |
| CVRPLTW | OR-Tools | | ✓ | ✓ | | | ✓ | ✓ | ✓ | ✓ | | ✓ | ✓ | |
| OCVRPBTW | OR-Tools | | ✓ | ✓ | | | ✓ | ✓ | ✓ | ✓ | ✓ | ✓ | ✓ | ✓ |
| CVRPBLTW | OR-Tools | | ✓ | ✓ | | | ✓ | ✓ | ✓ | ✓ | ✓ | ✓ | ✓ | |
| OCVRPL | OR-Tools | | ✓ | ✓ | | | | | | ✓ | | ✓ | ✓ | ✓ |
| CVRPBTW | OR-Tools | | ✓ | ✓ | | | ✓ | ✓ | ✓ | ✓ | ✓ | ✓ | ✓ | |
| OCVRPBL | OR-Tools | | ✓ | ✓ | | | | | | ✓ | ✓ | ✓ | ✓ | ✓ |
| OCVRPLTW | OR-Tools | | ✓ | ✓ | | | ✓ | ✓ | ✓ | ✓ | | ✓ | ✓ | ✓ |
| OCVRPBLTW | OR-Tools | | ✓ | ✓ | | | ✓ | ✓ | ✓ | ✓ | ✓ | ✓ | ✓ | ✓ |
| CVRPBP | OR-Tools | | ✓ | ✓ | | | | | | ✓ | ✓ | ✓ | ✓ | |
| OCVRPBP | OR-Tools | | ✓ | ✓ | | | | | | ✓ | ✓ | ✓ | ✓ | ✓ |
| CVRPBPL | OR-Tools | | ✓ | ✓ | | | | | | ✓ | ✓ | ✓ | ✓ | |
| OCVRPBPTW | OR-Tools | | ✓ | ✓ | | | ✓ | ✓ | ✓ | ✓ | ✓ | ✓ | ✓ | ✓ |
| CVRPBPLTW | OR-Tools | | ✓ | ✓ | | | ✓ | ✓ | ✓ | ✓ | ✓ | ✓ | ✓ | |
| CVRPBPTW | OR-Tools | | ✓ | ✓ | | | ✓ | ✓ | ✓ | ✓ | ✓ | ✓ | ✓ | |
| OCVRPBPL | OR-Tools | | ✓ | ✓ | | | | | | ✓ | ✓ | ✓ | ✓ | ✓ |
| OCVRPBPLTW | OR-Tools | | ✓ | ✓ | | | ✓ | ✓ | ✓ | ✓ | ✓ | ✓ | ✓ | ✓ |
| MDCVRP | PyVRP | | ✓ | ✓ | | | | | | ✓ | | ✓ | ✓ | |
| MDCVRPTW | PyVRP | | ✓ | ✓ | | | ✓ | ✓ | ✓ | ✓ | | ✓ | ✓ | |
| MDOCVRP | PyVRP | | ✓ | ✓ | | | | | | ✓ | | ✓ | ✓ | ✓ |
| MDCVRPL | PyVRP | | ✓ | ✓ | | | | | | ✓ | | ✓ | ✓ | |
| MDCVRPB | PyVRP | | ✓ | ✓ | | | | | | ✓ | ✓ | ✓ | ✓ | |
| MDOCVRPTW | PyVRP | | ✓ | ✓ | | | ✓ | ✓ | ✓ | ✓ | | ✓ | ✓ | ✓ |
| MDOCVRPB | PyVRP | | ✓ | ✓ | | | | | | ✓ | ✓ | ✓ | ✓ | ✓ |
| MDCVRPBL | PyVRP | | ✓ | ✓ | | | | | | ✓ | ✓ | ✓ | ✓ | |
| MDCVRPLTW | PyVRP | | ✓ | ✓ | | | ✓ | ✓ | ✓ | ✓ | | ✓ | ✓ | |
| MDOCVRPBTW | PyVRP | | ✓ | ✓ | | | ✓ | ✓ | ✓ | ✓ | ✓ | ✓ | ✓ | ✓ |
| MDCVRPBLTW | PyVRP | | ✓ | ✓ | | | ✓ | ✓ | ✓ | ✓ | ✓ | ✓ | ✓ | |
| MDOCVRPL | PyVRP | | ✓ | ✓ | | | | | | ✓ | | ✓ | ✓ | ✓ |
| MDCVRPBTW | PyVRP | | ✓ | ✓ | | | ✓ | ✓ | ✓ | ✓ | ✓ | ✓ | ✓ | |
| MDOCVRPBL | PyVRP | | ✓ | ✓ | | | | | | ✓ | ✓ | ✓ | ✓ | ✓ |
| MDOCVRPLTW | PyVRP | | ✓ | ✓ | | | ✓ | ✓ | ✓ | ✓ | | ✓ | ✓ | ✓ |
| MDOCVRPBLTW | PyVRP | | ✓ | ✓ | | | ✓ | ✓ | ✓ | ✓ | ✓ | ✓ | ✓ | ✓ |
| MDCVRPBP | PyVRP | | ✓ | ✓ | | | | | | ✓ | ✓ | ✓ | ✓ | |
| MDOCVRPBP | PyVRP | | ✓ | ✓ | | | | | | ✓ | ✓ | ✓ | ✓ | ✓ |
| MDCVRPBPL | PyVRP | | ✓ | ✓ | | | | | | ✓ | ✓ | ✓ | ✓ | |
| MDOCVRPBPTW | PyVRP | | ✓ | ✓ | | | ✓ | ✓ | ✓ | ✓ | ✓ | ✓ | ✓ | ✓ |
| MDCVRPBPLTW | PyVRP | | ✓ | ✓ | | | ✓ | ✓ | ✓ | ✓ | ✓ | ✓ | ✓ | |
| MDCVRPBPTW | PyVRP | | ✓ | ✓ | | | ✓ | ✓ | ✓ | ✓ | ✓ | ✓ | ✓ | |
| MDOCVRPBPL | PyVRP | | ✓ | ✓ | | | | | | ✓ | ✓ | ✓ | ✓ | ✓ |
| MDOCVRPBPLTW | PyVRP | | ✓ | ✓ | | | ✓ | ✓ | ✓ | ✓ | ✓ | ✓ | ✓ | ✓ |
| SPCTSP | ILS | | ✓ | | ✓ | ✓ | | | | ✓ | | | | |

*Table 12.* The problem representations of unseen 55 less-explored VRP variants. Here, RI represents the random identifier, and EAT, LAT, and ST indicate the earliest arrival time, the latest arrival time, and the service time, respectively.

| Problem | Oracle | RI | Coord. | Demand | Prize | Penalty | EAT | LAT | ST | Depot | Pickup | Delivery | Sub-routes | Open Route |
|---|---|---|---|---|---|---|---|---|---|---|---|---|---|---|
| ACVRPTW | OR-Tools | ✓ | | ✓ | | | ✓ | ✓ | ✓ | ✓ | | ✓ | ✓ | |
| AOCVRP | OR-Tools | ✓ | | ✓ | | | | | | ✓ | | ✓ | ✓ | ✓ |
| ACVRPL | OR-Tools | ✓ | | ✓ | | | | | | ✓ | | ✓ | ✓ | |
| ACVRPB | OR-Tools | ✓ | | ✓ | | | | | | ✓ | ✓ | ✓ | ✓ | |
| AOCVRPTW | OR-Tools | ✓ | | ✓ | | | ✓ | ✓ | ✓ | ✓ | | ✓ | ✓ | ✓ |
| AOCVRPB | OR-Tools | ✓ | | ✓ | | | | | | ✓ | ✓ | ✓ | ✓ | ✓ |
| ACVRPBL | OR-Tools | ✓ | | ✓ | | | | | | ✓ | ✓ | ✓ | ✓ | |
| ACVRPLTW | OR-Tools | ✓ | | ✓ | | | ✓ | ✓ | ✓ | ✓ | | ✓ | ✓ | |
| AOCVRPBTW | OR-Tools | ✓ | | ✓ | | | ✓ | ✓ | ✓ | ✓ | ✓ | ✓ | ✓ | ✓ |
| ACVRPBLTW | OR-Tools | ✓ | | ✓ | | | ✓ | ✓ | ✓ | ✓ | ✓ | ✓ | ✓ | |
| AOCVRPL | OR-Tools | ✓ | | ✓ | | | | | | ✓ | | ✓ | ✓ | ✓ |
| ACVRPBTW | OR-Tools | ✓ | | ✓ | | | ✓ | ✓ | ✓ | ✓ | ✓ | ✓ | ✓ | |
| AOCVRPBL | OR-Tools | ✓ | | ✓ | | | | | | ✓ | ✓ | ✓ | ✓ | ✓ |
| AOCVRPLTW | OR-Tools | ✓ | | ✓ | | | ✓ | ✓ | ✓ | ✓ | | ✓ | ✓ | ✓ |
| AOCVRPBLTW | OR-Tools | ✓ | | ✓ | | | ✓ | ✓ | ✓ | ✓ | ✓ | ✓ | ✓ | ✓ |
| ACVRPBP | OR-Tools | ✓ | | ✓ | | | | | | ✓ | ✓ | ✓ | ✓ | |
| AOCVRPBP | OR-Tools | ✓ | | ✓ | | | | | | ✓ | ✓ | ✓ | ✓ | ✓ |
| ACVRPBPL | OR-Tools | ✓ | | ✓ | | | | | | ✓ | ✓ | ✓ | ✓ | |
| AOCVRPBPTW | OR-Tools | ✓ | | ✓ | | | ✓ | ✓ | ✓ | ✓ | ✓ | ✓ | ✓ | ✓ |
| ACVRPBPLTW | OR-Tools | ✓ | | ✓ | | | ✓ | ✓ | ✓ | ✓ | ✓ | ✓ | ✓ | |
| ACVRPBPTW | OR-Tools | ✓ | | ✓ | | | ✓ | ✓ | ✓ | ✓ | ✓ | ✓ | ✓ | |
| AOCVRPBPL | OR-Tools | ✓ | | ✓ | | | | | | ✓ | ✓ | ✓ | ✓ | ✓ |
| AOCVRPBPLTW | OR-Tools | ✓ | | ✓ | | | ✓ | ✓ | ✓ | ✓ | ✓ | ✓ | ✓ | ✓ |
| AMDCVRP | OR-Tools | ✓ | | ✓ | | | | | | ✓ | | ✓ | ✓ | |
| AMDCVRPTW | OR-Tools | ✓ | | ✓ | | | ✓ | ✓ | ✓ | ✓ | | ✓ | ✓ | |
| AMDOCVRP | OR-Tools | ✓ | | ✓ | | | | | | ✓ | | ✓ | ✓ | ✓ |
| AMDCVRPL | OR-Tools | ✓ | | ✓ | | | | | | ✓ | | ✓ | ✓ | |
| AMDCVRPB | OR-Tools | ✓ | | ✓ | | | | | | ✓ | ✓ | ✓ | ✓ | |
| AMDOCVRPTW | OR-Tools | ✓ | | ✓ | | | ✓ | ✓ | ✓ | ✓ | | ✓ | ✓ | ✓ |
| AMDOCVRPB | OR-Tools | ✓ | | ✓ | | | | | | ✓ | ✓ | ✓ | ✓ | ✓ |
| AMDCVRPBL | OR-Tools | ✓ | | ✓ | | | | | | ✓ | ✓ | ✓ | ✓ | |
| AMDCVRPLTW | OR-Tools | ✓ | | ✓ | | | ✓ | ✓ | ✓ | ✓ | | ✓ | ✓ | |
| AMDOCVRPBTW | OR-Tools | ✓ | | ✓ | | | ✓ | ✓ | ✓ | ✓ | ✓ | ✓ | ✓ | ✓ |
| AMDCVRPBLTW | OR-Tools | ✓ | | ✓ | | | ✓ | ✓ | ✓ | ✓ | ✓ | ✓ | ✓ | |
| AMDOCVRPL | OR-Tools | ✓ | | ✓ | | | | | | ✓ | | ✓ | ✓ | ✓ |
| AMDCVRPBTW | OR-Tools | ✓ | | ✓ | | | ✓ | ✓ | ✓ | ✓ | ✓ | ✓ | ✓ | |
| AMDOCVRPBL | OR-Tools | ✓ | | ✓ | | | | | | ✓ | ✓ | ✓ | ✓ | ✓ |
| AMDOCVRPLTW | OR-Tools | ✓ | | ✓ | | | ✓ | ✓ | ✓ | ✓ | | ✓ | ✓ | ✓ |
| AMDOCVRPBLTW | OR-Tools | ✓ | | ✓ | | | ✓ | ✓ | ✓ | ✓ | ✓ | ✓ | ✓ | ✓ |
| AMDCVRPBP | OR-Tools | ✓ | | ✓ | | | | | | ✓ | ✓ | ✓ | ✓ | |
| AMDOCVRPBP | OR-Tools | ✓ | | ✓ | | | | | | ✓ | ✓ | ✓ | ✓ | ✓ |
| AMDCVRPBPL | OR-Tools | ✓ | | ✓ | | | | | | ✓ | ✓ | ✓ | ✓ | |
| AMDOCVRPBPTW | OR-Tools | ✓ | | ✓ | | | ✓ | ✓ | ✓ | ✓ | ✓ | ✓ | ✓ | ✓ |
| AMDCVRPBPLTW | OR-Tools | ✓ | | ✓ | | | ✓ | ✓ | ✓ | ✓ | ✓ | ✓ | ✓ | |
| AMDCVRPBPTW | OR-Tools | ✓ | | ✓ | | | ✓ | ✓ | ✓ | ✓ | ✓ | ✓ | ✓ | |
| AMDOCVRPBPL | OR-Tools | ✓ | | ✓ | | | | | | ✓ | ✓ | ✓ | ✓ | ✓ |
| AMDOCVRPBPLTW | OR-Tools | ✓ | | ✓ | | | ✓ | ✓ | ✓ | ✓ | ✓ | ✓ | ✓ | ✓ |
| APDTSP | OR-Tools | ✓ | | | | | | | | ✓ | ✓ | ✓ | | |
| APDCVRP | OR-Tools | ✓ | | ✓ | | | | | | ✓ | ✓ | ✓ | ✓ | |
| AOPDCVRP | OR-Tools | ✓ | | ✓ | | | | | | ✓ | ✓ | ✓ | ✓ | ✓ |
| PDCVRP | OR-Tools | | ✓ | ✓ | | | | | | ✓ | ✓ | ✓ | ✓ | |
| OPDCVRP | OR-Tools | | ✓ | ✓ | | | | | | ✓ | ✓ | ✓ | ✓ | ✓ |
| AOP | OR-Tools | ✓ | | | ✓ | | | | | ✓ | | | | |
| APCTSP | OR-Tools | ✓ | | | ✓ | ✓ | | | | ✓ | | | | |
| ASPCTSP | N/A | ✓ | | | ✓ | ✓ | | | | ✓ | | | | |

# E. Model Architecture and Training

### E.1. Encoder

Similar to previous work (Kool et al., 2019; Kwon et al., 2020), we also use the encoder composed of $L$ stacked attention layers, where each attention layer consists of two sub-layers: an attention sub-layer and a Feed-Forward (FF) sub-layer, both of which use Instance Normalization (Ulyanov et al., 2016) and skip-connection (He et al., 2016).

The encoder $\boldsymbol{\theta}_{enc}$ with $L$ stacked attention layers transforms $H^{(0)}$ into advanced node embeddings $H^{(L)} = \{\mathbf{h}_i^{(L)}\}_{i=0}^n$. Let $H^{(\ell-1)} = \{\mathbf{h}_i^{(\ell-1)}\}_{i=0}^n$ denote the input to the $\ell$-th attention layer for $\ell = 1, \ldots, L$. The outputs for the $i$-th node are computed as follows:

$$\widetilde{\mathbf{h}}_i^{(\ell)} = \mathrm{IN}^{(\ell)} \left( \mathbf{h}_i^{(\ell-1)} + \mathrm{MBM}^{(\ell)} \left( \mathbf{h}_i^{(\ell-1)}, H^{(\ell-1)}, \boldsymbol{D}, \boldsymbol{D}^{\mathrm{T}}, \boldsymbol{R} \right) \right), \tag{11}$$

$$\mathbf{h}_i^{(\ell)} = \mathrm{IN}^{(\ell)} \left( \widetilde{\mathbf{h}}_i^{(\ell)} + \mathrm{FF}^{(\ell)} \left( \widetilde{\mathbf{h}}_i^{(\ell)} \right) \right), \tag{12}$$

where $\mathrm{IN}(\cdot)$ denotes instance normalization (Ulyanov et al., 2016), MBM represents the adopted mixed bias module (see Equation (5) for details), and $\mathrm{FF}(\cdot)$ in Equation (12) corresponds to a fully connected neural network with ReLU activation. After $L$ attention layers, the final node embeddings $H^{(L)} = \{\mathbf{h}_i^{(L)}\}_{i=0}^n$ encapsulate the advanced feature representations of each node. Notably, to ensure dimensional consistency across bias channels, we normalize $\boldsymbol{D}$ and $\boldsymbol{D}^{\mathbf{T}}$ by dividing their maximum value before passing the MBM.

### E.2. Decoder

In URS, all parameters $\boldsymbol{\theta}_{dec}(\boldsymbol{\lambda})$ are adaptively generated conditioned on the multi-hot problem representation $\boldsymbol{\lambda}$ (see Appendix D for detailed problem representations). $\boldsymbol{\theta}_{dec}(\boldsymbol{\lambda})$ includes two MLP networks, which are $\mathrm{WEIGHT}(\boldsymbol{\lambda})$ and $\mathrm{BIAS}(\boldsymbol{\lambda})$. $\mathrm{WEIGHT}(\boldsymbol{\lambda})$ is used to generate linear projection matrices $[W_Q(\boldsymbol{\lambda}), W_K(\boldsymbol{\lambda}), W_V(\boldsymbol{\lambda})]$. $\mathrm{BIAS}(\boldsymbol{\lambda})$ in Equation (6) is used to generate bias weights for $\mathrm{Attention}(\cdot)$(see Equation (9)) and compatibility (see Equation (10)).

For $\mathrm{WEIGHT}(\boldsymbol{\lambda})$, we use a multi-layer MLP hypernetwork conditioned on a 13-dimensional problem representation $\boldsymbol{\lambda}$ to generate problem-specific adaptive parameters. Note that we factor $W_Q(\boldsymbol{\lambda}) \in \mathbb{R}^{(2d+1) \times d}$ into three vertically concatenated blocks $W_{\mathrm{first}}(\boldsymbol{\lambda}) \in \mathbb{R}^{d \times d}$, $W_{\mathrm{last}}(\boldsymbol{\lambda}) \in \mathbb{R}^{d \times d}$ and $W_C(\boldsymbol{\lambda}) \in \mathbb{R}^{1 \times d}$, applied respectively to the first visited node $\mathbf{h}_{\pi_1}^{(L)}$, the last visited node $\mathbf{h}_{\pi_{t-1}}^{(L)}$, and a scalar state feature $C_t \in \mathbb{R}^1$ (see Appendix E.3 for more details). Thus $\mathrm{WEIGHT}(\boldsymbol{\lambda})$ outputs the linear projection matrices $\{W_{\mathrm{first}}(\boldsymbol{\lambda}), W_{\mathrm{last}}(\boldsymbol{\lambda}), W_C(\boldsymbol{\lambda}), W_K(\boldsymbol{\lambda}), W_V(\boldsymbol{\lambda})\}$.

Furthermore, we adopt the parameter compression technique of Ha et al. (2016) to control model size. Specifically, we first generate an advanced embedding matrix $\boldsymbol{H}_{\mathrm{dec}}(\boldsymbol{\lambda}) = [\boldsymbol{h}_{\mathrm{dec}}^i(\boldsymbol{\lambda})|i = 1, 2, \ldots, 5] \in \mathbb{R}^{5 \times |\boldsymbol{\lambda}|}$ whose five row embeddings each parameterize one target projection matrix. It can be computed via three consecutive linear projections:

$$\boldsymbol{H}_{\mathrm{dec}}(\boldsymbol{\lambda}) = ((\boldsymbol{\lambda} W_1 + \mathbf{b}_1) W_2 + \mathbf{b}_2) W_3 + \mathbf{b}_3, \tag{13}$$

where $W_1 \in \mathbb{R}^{|\boldsymbol{\lambda}| \times d_h}$, $W_2 \in \mathbb{R}^{d_h \times d_h}$, $W_3 \in \mathbb{R}^{d_h \times (5 \times |\boldsymbol{\lambda}|)}$, $\mathbf{b}_1 \in \mathbb{R}^{d_h}$, $\mathbf{b}_2 \in \mathbb{R}^{d_h}, \mathbf{b}_3 \in \mathbb{R}^{(5 \times |\boldsymbol{\lambda}|)}$ are learnable parameters, $d_h$ is 256 in this work, following Lin et al. (2022). Next, we use five different linear projections to map the new representation $\boldsymbol{h}_{\mathrm{dec}}^i(\boldsymbol{\lambda})$ of each row to the corresponding decoder parameters. The computation can be expressed as:

$$W_{\mathrm{first}}(\boldsymbol{\lambda}) = \boldsymbol{h}_{\mathrm{dec}}^1(\boldsymbol{\lambda}) W_{\mathbf{hyper}}^1, \; W_{\mathrm{last}}(\boldsymbol{\lambda}) = \boldsymbol{h}_{\mathrm{dec}}^2(\boldsymbol{\lambda}) W_{\mathbf{hyper}}^2, \; W_C(\boldsymbol{\lambda}) = \boldsymbol{h}_{\mathrm{dec}}^3(\boldsymbol{\lambda}) W_{\mathbf{hyper}}^3, \tag{14}$$

$$W_K(\boldsymbol{\lambda}) = \boldsymbol{h}_{\mathrm{dec}}^4(\boldsymbol{\lambda}) W_{\mathbf{hyper}}^4, \; W_V(\boldsymbol{\lambda}) = \boldsymbol{h}_{\mathrm{dec}}^5(\boldsymbol{\lambda}) W_{\mathbf{hyper}}^5, \tag{15}$$

where $W_{\mathbf{hyper}}^1 \in \mathbb{R}^{|\boldsymbol{\lambda}| \times (d \times d)}$, $W_{\mathbf{hyper}}^2 \in \mathbb{R}^{|\boldsymbol{\lambda}| \times (d \times d)}$, $W_{\mathbf{hyper}}^3 \in \mathbb{R}^{|\boldsymbol{\lambda}| \times (1 \times d)}$, $W_{\mathbf{hyper}}^4 \in \mathbb{R}^{|\boldsymbol{\lambda}| \times (d \times d)}$, $W_{\mathbf{hyper}}^5 \in \mathbb{R}^{|\boldsymbol{\lambda}| \times (d \times d)}$ are learnable parameters. In this way, we use a simple MLP hypernetwork $\mathrm{WEIGHT}(\boldsymbol{\lambda})$ to generate used linear projection matrices $\{W_Q(\boldsymbol{\lambda}), W_K(\boldsymbol{\lambda}), W_V(\boldsymbol{\lambda})\}$ conditioned on the multi-hot problem representation $\boldsymbol{\lambda}$, enabling the adaptive construction of high-quality solutions for different problems.

### E.3. Problem State $C_t$

In this section, we detail the problems that require additional state signals. If a problem is not mentioned in this section, it means we will not apply any extra state to it, i.e., $C_t = 0$.

For all problems with the capacity constraint, we explicitly impose **only remaining load**, while diverse additional constraints (e.g., time windows, duration limit, and backhaul) are enforced implicitly as $\mathcal{M}_t$ masks infeasible candidate nodes to guarantee valid solutions, instead of being fully enumerated in the input of the decoder. For OP and AOP, we keep track of the remaining maximum length at time $t$, and $C_t$ represents the remaining routing length that can be moved, and we normalize it to $[0, 1]$ by dividing by the maximum length (i.e., 4.0 in OP100 and 1.0 in AOP100). The setting is the same as AM (Kool et al., 2019). For (A)PCTSP and (A)SPCTSP, $C_t$ is the remaining prize to collect, and we do not provide any information about the penalties, as this is irrelevant for the remaining decisions, following Kool et al. (2019). For the problem state, we provide an ablation study in Appendix K.2.

### E.4. Training

Following Kwon et al. (2020), we use $n$ trajectories with distinct starting nodes in training. URS is trained by the REINFORCE (Williams, 1992) algorithm with a shared baseline (Kwon et al., 2020):

$$\nabla_\theta \mathcal{L}(\boldsymbol{\theta}) = \mathbb{E}_{p(\boldsymbol{\pi}|\mathcal{G},\boldsymbol{\theta})}[(f(\boldsymbol{\pi}|\mathcal{G}) - \frac{1}{n}\sum_{i=1}^{n} f(\boldsymbol{\pi}^i|\mathcal{G}))\nabla_\theta \log p_\theta(\boldsymbol{\pi}|\mathcal{G})], \tag{16}$$

$$p_\theta(\boldsymbol{\pi} \mid \mathcal{G}) = \prod_{t=2}^{n} p_\theta(\pi_t \mid \mathcal{G}, \pi_{1:t-1}), \tag{17}$$

where the objective function $f(\boldsymbol{\pi}|\mathcal{G})$ represents the total reward (e.g., the negative value of tour length) of instance $\mathcal{G}$ given a specific solution $\boldsymbol{\pi}$.

## F. Adaptation Attention Free Module

Following Zhou et al. (2026), we implement $\text{Attention}(\cdot)$ using an adaptation attention free module (AAFM) to enhance geometric pattern recognition for routing problems. Given the input $X$, the AAFM computation is then expressed as:

$$Q = XW^Q, \quad K = XW^K, \quad V = XW^V, \tag{18}$$

$$\text{Attention}(Q, K, V, A) = \sigma(Q) \odot \frac{\exp(A)(\exp(K) \odot V)}{\exp(A)\exp(K)}, \tag{19}$$

where $W^Q$, $W^K$, and $W^V$ are learnable matrices. $\sigma$ denotes the sigmoid function, $\odot$ represents the element-wise product, and $A = \{a_{ij}\}$ denotes the pair-wise adaptation bias. For distance matrices, the corresponding adaptation bias $f(\alpha, N, d_{ij}) = -\alpha \cdot \log_2 N \cdot d_{i,j}$. Here, $N$ denotes the total number of nodes (i.e., problem size), and $\alpha$ is generated via a lightweight network $\text{BIAS}(\boldsymbol{\lambda})$ in our paper (see Equation (6)). Notably, for the relation matrix, we remove the scale $N$ in the calculation of adaptation bias because the relation is scale-independent. Compared to multi-head attention (MHA) (Vaswani et al., 2017), AAFM enables the model to explicitly capture instance-specific knowledge by updating pair-wise adaptation biases while exhibiting lower computational overhead. Further details are provided in the related work mentioned above.

## G. Comparison between Different Attention Mechanisms

*Table 13.* Comparison between different attention mechanisms. Here, $N, d, m$ denote the sequence length, feature dimension, and extra hidden layer dimension of mixed-score attention, respectively. Values in parentheses represent the percentage increase or decrease relative to URS and Avg. Peak VRAM represents the average peak VRAM per instance.

| Attention Mechanism | Time Complexity | Space Complexity | #Params | Avg.Peak VRAM | Inference Time |
|---|---|---|---|---|---|
| Multi-Head Attention | $\mathcal{O}(N^2 d)$ | $\mathcal{O}(N^2 + Nd)$ | 3.32M($\times$0.7) | 1.87MB($\times$0.99) | 0.92s($\times$0.9) |
| Mixed-Score Attention (Parallel) | $\mathcal{O}(N^2(d+m))$ | $\mathcal{O}(N^2 m + Nd)$ | 16.17M($\times$3.3) | 25.64MB($\times$13.6) | 3.64s($\times$3.7) |
| Mixed Bias Module (Ours) | $\mathcal{O}(N^2 d)$ | $\mathcal{O}(N^2 + Nd)$ | 4.96M | 1.89MB | $0.97s$ |

To validate the efficiency of our proposed MBM, we have conducted a new comparative experiment between different attentions and tested them on TSP instances to ensure consistent decoding steps. Specifically, we replace our MBM with both a vanilla MHA (Vaswani et al., 2017) and the parallel mixed-score attention (MSA) used in MatNet (Kwon et al., 2021),

and compare their complexities, memory overheads, and inference time. The results in Table 13 show that the increased VRAM and time overhead of MBM compared to standard MHA are small, and both maintain the same time complexity and space complexity. Compared to MatNet's parallel MSA, our MBM drastically reduces both VRAM consumption and inference latency. These results show that MBM imposes only modest computational penalties while successfully capturing crucial problem-specific biases, which enhances cross-problem generalization while reducing architectural redundancy.

## H. Training and Model Settings

### H.1. Multi-Task Version

The detailed information about the hyperparameter settings can be found in Table 14. The URS training is conducted on a single NVIDIA GeForce RTX 4090 GPU (24GB of memory). The training procedure requires only about 95 hours ($\approx$ 4 days) in total (500 epochs at approximately 12 minutes each) to obtain a unified parameter set capable of addressing a broad class of routing problems.

### H.2. Single-Task Version

URS-STL is trained on data from only a single problem type (e.g., "URS-STL (CVRP)" is trained only on CVRP). The URS-STL variant uses exactly the same model architecture, hyperparameters (e.g., embedding dimension, number of layers, optimizer), and training setup (e.g., initial learning rate, batch size) as the multi-task URS described in Section 4.1 and Appendix H.1. The only modification we made is to the training duration. We observe that the single-task models converge significantly faster than the multi-task model. Therefore, we train URS-STL models for 300 epochs, with a learning rate decay applied at the 251st epoch. This contrasts with the multi-task URS, which is trained for 500 epochs with decay applied at the 451st epoch. This adjustment is made solely to reflect the faster convergence of the single-task training. All other hyperparameters and model architecture remain identical.

*Table 14.* The hyperparameter settings of URS.

| Hyperparameter | Value |
|---|---|
| Optimizer | AdamW |
| Clipping parameter $\zeta$ | 50 |
| Initial learning rate | $10^{-4}$ |
| Epochs of learning rate decay | [451] |
| Factor of learning rate decay | 0.1 |
| Weight decay | $10^{-6}$ |
| The number of encoder layers $L$ | 12 |
| Embedding dimension $d$ | 128 |
| Feed forward dimension | 512 |
| Dimension $d_h$ in Equation (13) | 256 |
| Training scale | 100 |
| Batches of each epoch | 2,000 |
| Batch size | 128 |
| Training Epochs | 500 |

## I. Generalization to Unseen VRPs

To comprehensively evaluate URS's zero-shot generalization, we test its performance on a diverse set of 99 unseen problems. This set is designed to cover both established and novel challenges, including 44 well-studied VRP variants and 55 less-explored VRP variants. The detailed results are presented in Table 15 and Table 16. On widely-studied constraints (i.e., C, O, L, TW, MD, B, BP), URS continues to deliver high-quality solutions. Notably, URS merely selects the next node from a given set of feasible candidates without any additional domain knowledge, making its performance especially meaningful. Benefiting from our UDR, URS can directly address 55 less-explored VRP variants that compound asymmetry and PD relations. Without any fine-tuning, URS still produces high-quality solutions for them, thereby strengthening its broad problem generality and deployment efficiency.

*Table 15.* Zero-shot generalization performance across 44 unseen well-studied VRP variants ($N = 100$).

| Method | CVRPBP | | OCVRPBP | | CVRPBPL | | OCVRPBPTW | | CVRPBPLTW | |
|---|---|---|---|---|---|---|---|---|---|---|
| | Gap | Time | Gap | Time | Gap | Time | Gap | Time | Gap | Time |
| Oracle | 0.00% | 6.6m | 0.00% | 6.6m | 0.00% | 6.6m | 0.00% | 6.6m | 0.00% | 6.6m |
| MVMoE | 13.95% | 14s | 18.27% | 14s | 13.38% | 14s | 9.42% | 15s | 16.66% | 16s |
| MTPOMO | 13.71% | 7s | 17.90% | 8s | 13.44% | 8s | 9.94% | 9s | 17.15% | 10s |
| ReLD-MTL | 13.57% | 9s | **15.81%** | 9s | 12.96% | 10s | **8.43%** | 11s | **15.95%** | 12s |
| URS | **12.95%** | 7s | 16.96% | 7s | **12.65%** | 8s | 10.05% | 8s | 18.33% | 10s |

| Method | CVRPBPTW | | OCVRPBPL | | OCVRPBPLTW | | CVRPL | | CVRPBL | |
|---|---|---|---|---|---|---|---|---|---|---|
| | Gap | Time | Gap | Time | Gap | Time | Gap | Time | Gap | Time |
| Oracle | 0.00% | 6.6m | 0.00% | 6.6m | 0.00% | 6.6m | 0.00% | 16m | 0.00% | 3.5h |
| MVMoE | 16.86% | 15s | 17.75% | 14s | 8.87% | 16s | 0.26% | 12s | 1.35% | 12s |
| MTPOMO | 17.22% | 9s | 17.31% | 8s | 9.47% | 10s | 0.48% | 6s | 1.79% | 6s |
| ReLD-MTL | **16.09%** | 11s | **15.21%** | 10s | **8.06%** | 11s | **0.02%** | 8s | **1.01%** | 7s |
| URS | 18.15% | 9s | 16.37% | 8s | 9.51% | 9s | 0.43% | 7s | 1.65% | 6s |

| Method | CVRPBTW | | OCVRPBL | | OCVRPLTW | | OCVRPBLTW | | MDCVRPBP | |
|---|---|---|---|---|---|---|---|---|---|---|
| | Gap | Time | Gap | Time | Gap | Time | Gap | Time | Gap | Time |
| Oracle | 0.00% | 3.5h | 0.00% | 3.5h | 0.00% | 3.5h | 0.00% | 3.5h | 0.00% | 6.6m |
| MVMoE | 7.08% | 15s | 7.12% | 12s | 3.90% | 15s | 10.01% | 15s | 65.98% | 33s |
| MTPOMO | 7.41% | 7s | 7.34% | 7s | 4.37% | 8s | 10.50% | 7s | 45.77% | 24s |
| ReLD-MTL | **6.74%** | 8s | **5.41%** | 7s | **3.16%** | 9s | **9.22%** | 8s | 40.70% | 27s |
| URS | 8.94% | 8s | 9.47% | 7s | 5.12% | 9s | 13.72% | 8s | **35.44%** | 24s |

| Method | MDOCVRPBP | | MDCVRPBPL | | MDOCVRPBPTW | | MDCVRPBPLTW | | MDCVRPBPTW | |
|---|---|---|---|---|---|---|---|---|---|---|
| | Gap | Time | Gap | Time | Gap | Time | Gap | Time | Gap | Time |
| Oracle | 0.00% | 6.6m | 0.00% | 6.6m | 0.00% | 6.6m | 0.00% | 6.6m | 0.00% | 6.6m |
| MVMoE | 56.60% | 33s | 65.81% | 36s | 63.77% | 38s | 68.49% | 42s | 69.00% | 39s |
| MTPOMO | 42.94% | 24s | 45.98% | 26s | 47.09% | 29s | 50.45% | 32s | 50.74% | 30s |
| ReLD-MTL | 37.74% | 27s | 39.57% | 29s | 45.16% | 34s | 45.23% | 37s | 45.62% | 35s |
| URS | **24.44%** | 22s | **35.07%** | 26s | **26.31%** | 26s | **36.49%** | 32s | **36.71%** | 30s |

| Method | MDOCVRPBPL | | MDOCVRPBPLTW | | MDCVRP | | MDCVRPTW | | MDOCVRP | |
|---|---|---|---|---|---|---|---|---|---|---|
| | Gap | Time | Gap | Time | Gap | Time | Gap | Time | Gap | Time |
| Oracle | 0.00% | 6.6m | 0.00% | 6.6m | 0.00% | 6.6m | 0.00% | 6.6m | 0.00% | 6.6m |
| MVMoE | 56.26% | 34s | 62.47% | 40s | 33.06% | 31s | 53.64% | 41s | 29.08% | 31s |
| MTPOMO | 41.98% | 27s | 46.26% | 32s | 23.38% | 23s | 36.10% | 31s | 27.46% | 23s |
| ReLD-MTL | 37.45% | 29s | 44.74% | 36s | 18.39% | 25s | **30.90%** | 34s | 22.32% | 25s |
| URS | **24.22%** | 24s | **26.20%** | 28s | **15.05%** | 23s | 37.26% | 32s | **22.01%** | 23s |

| Method | MDCVRPL | | MDCVRPB | | MDOCVRPTW | | MDOCVRPB | | MDCVRPBL | |
|---|---|---|---|---|---|---|---|---|---|---|
| | Gap | Time | Gap | Time | Gap | Time | Gap | Time | Gap | Time |
| Oracle | 0.00% | 6.6m | 0.00% | 6.6m | 0.00% | 6.6m | 0.00% | 6.6m | 0.00% | 6.6m |
| MVMoE | 33.42% | 34s | 21.70% | 26s | 51.57% | 40s | 28.21% | 26s | 20.35% | 29s |
| MTPOMO | 24.32% | 25s | 12.89% | 19s | 35.71% | 29s | 25.06% | 20s | 11.10% | 20s |
| ReLD-MTL | 18.48% | 28s | **9.38%** | 21s | 33.55% | 33s | 18.50% | 21s | **8.26%** | 23s |
| URS | **15.32%** | 26s | 13.14% | 19s | **26.05%** | 25s | **15.17%** | 18s | 11.19% | 21s |

| Method | MDCVRPLTW | | MDOCVRPBTW | | MDCVRPBLTW | | MDOCVRPL | | MDCVRPBTW | |
|---|---|---|---|---|---|---|---|---|---|---|
| | Gap | Time | Gap | Time | Gap | Time | Gap | Time | Gap | Time |
| Oracle | 0.00% | 6.6m | 0.00% | 6.6m | 0.00% | 6.6m | 0.00% | 6.6m | 0.00% | 6.6m |
| MVMoE | 55.55% | 44s | 62.47% | 35s | 63.18% | 37s | 29.33% | 33s | 65.90% | 35s |
| MTPOMO | 37.34% | 32s | 45.55% | 25s | 45.07% | 27s | 27.21% | 25s | 45.94% | 25s |
| ReLD-MTL | **31.56%** | 36s | 43.36% | 28s | 40.57% | 30s | **22.05%** | 27s | 41.28% | 28s |
| URS | 38.04% | 35s | **30.32%** | 22s | **37.70%** | 27s | 22.15% | 25s | **37.78%** | 25s |

| Method | MDOCVRPBL | | MDOCVRPLTW | | MDOCVRPBLTW | | SPCTSP | | — | |
|---|---|---|---|---|---|---|---|---|---|---|
| | Gap | Time | Gap | Time | Gap | Time | Gap | Time | | |
| Oracle | 0.00% | 6.6m | 0.00% | 6.6m | 0.00% | 6.6m | 0.00% | 1.6h | — | |
| MVMoE | 28.70% | 28s | 51.09% | 43s | 62.10% | 37s | — | — | — | |
| MTPOMO | 25.01% | 21s | 35.39% | 31s | 45.48% | 27s | — | — | — | |
| ReLD-MTL | 18.50% | 22s | 33.26% | 35s | 43.27% | 30s | — | — | — | |
| URS | **14.90%** | 20s | **25.59%** | 27s | **30.41%** | 24s | **-2.37%** | 5s | — | |

*Table 16.* Zero-shot generalization performance across 55 unseen less-explored VRP variants ($N = 100$).

| Method | ACVRPL Gap | ACVRPL Time | ACVRPBL Gap | ACVRPBL Time | ACVRPLTW Gap | ACVRPLTW Time | ACVRPBLTW Gap | ACVRPBLTW Time | AOCVRPL Gap | AOCVRPL Time |
|---|---|---|---|---|---|---|---|---|---|---|
| Oracle | 0.00% | 6.6m | 0.00% | 6.6m | 0.00% | 6.6m | 0.00% | 6.6m | 0.00% | 6.6m |
| URS | -2.24% | 1.9m | 6.18% | 2.4m | 12.22% | 2.7m | 10.18% | 2.4m | 37.36% | 3.1m |

| Method | AOCVRPBL Gap | AOCVRPBL Time | AOCVRPLTW Gap | AOCVRPLTW Time | AOCVRPBLTW Gap | AOCVRPBLTW Time | ACVRPBPL Gap | ACVRPBPL Time | ACVRPBPLTW Gap | ACVRPBPLTW Time |
|---|---|---|---|---|---|---|---|---|---|---|
| Oracle | 0.00% | 6.6m | 0.00% | 6.6m | 0.00% | 6.6m | 0.00% | 6.6m | 0.00% | 6.6m |
| URS | 17.23% | 2.7m | 16.35% | 2.8m | 14.32% | 2.5m | 24.03% | 2.6m | 13.82% | 2.8m |

| Method | AOCVRPBPL Gap | AOCVRPBPL Time | AOCVRPBPLTW Gap | AOCVRPBPLTW Time | ACVRPB Gap | ACVRPB Time | ACVRPTW Gap | ACVRPTW Time | AOCVRP Gap | AOCVRP Time |
|---|---|---|---|---|---|---|---|---|---|---|
| Oracle | 0.00% | 6.6m | 0.00% | 6.6m | 0.00% | 6.6m | 0.00% | 6.6m | 0.00% | 6.6m |
| URS | 32.50% | 2.7m | 16.43% | 2.7m | 7.10% | 2.1m | 11.28% | 2.5m | 37.63% | 2.9m |

| Method | AOCVRPTW Gap | AOCVRPTW Time | AOCVRPB Gap | AOCVRPB Time | AOCVRPBTW Gap | AOCVRPBTW Time | ACVRPBTW Gap | ACVRPBTW Time | ACVRPBP Gap | ACVRPBP Time |
|---|---|---|---|---|---|---|---|---|---|---|
| Oracle | 0.00% | 6.6m | 0.00% | 6.6m | 0.00% | 6.6m | 0.00% | 6.6m | 0.00% | 6.6m |
| URS | 17.47% | 2.6m | 18.05% | 2.5m | 14.45% | 2.3m | 9.95% | 2.2m | 24.20% | 2.4m |

| Method | AOCVRPBP Gap | AOCVRPBP Time | AOCVRPBPTW Gap | AOCVRPBPTW Time | ACVRPBPTW Gap | ACVRPBPTW Time | APDTSP Gap | APDTSP Time | AMDCVRPL Gap | AMDCVRPL Time |
|---|---|---|---|---|---|---|---|---|---|---|
| Oracle | 0.00% | 6.6m | 0.00% | 6.6m | 0.00% | 6.6m | 0.00% | 6.6m | 0.00% | 6.6m |
| URS | 32.58% | 2.5m | 17.45% | 2.5m | 14.52% | 2.6m | 6.21% | 1m | 7.15% | 6.1m |

| Method | AMDCVRPBL Gap | AMDCVRPBL Time | AMDCVRPLTW Gap | AMDCVRPLTW Time | AMDCVRPBLTW Gap | AMDCVRPBLTW Time | AMDOCVRPL Gap | AMDOCVRPL Time | AMDOCVRPBL Gap | AMDOCVRPBL Time |
|---|---|---|---|---|---|---|---|---|---|---|
| Oracle | 0.00% | 6.6m | 0.00% | 6.6m | 0.00% | 6.6m | 0.00% | 6.6m | 0.00% | 6.6m |
| URS | 17.51% | 6.5m | 19.98% | 8.8m | 16.73% | 7.5m | 47.80% | 8.6m | 27.23% | 7.1m |

| Method | AMDOCVRPLTW Gap | AMDOCVRPLTW Time | AMDOCVRPBLTW Gap | AMDOCVRPBLTW Time | AMDCVRPBPL Gap | AMDCVRPBPL Time | AMDCVRPBPLTW Gap | AMDCVRPBPLTW Time | AMDOCVRPBPL Gap | AMDOCVRPBPL Time |
|---|---|---|---|---|---|---|---|---|---|---|
| Oracle | 0.00% | 6.6m | 0.00% | 6.6m | 0.00% | 6.6m | 0.00% | 6.6m | 0.00% | 6.6m |
| URS | 24.90% | 7.9m | 22.84% | 6.7m | 35.00% | 7.2m | 20.50% | 8.3m | 38.78% | 7.3m |

| Method | AMDOCVRPBPLTW Gap | AMDOCVRPBPLTW Time | AMDCVRP Gap | AMDCVRP Time | AMDCVRPTW Gap | AMDCVRPTW Time | AMDOCVRP Gap | AMDOCVRP Time | AMDCVRPB Gap | AMDCVRPB Time |
|---|---|---|---|---|---|---|---|---|---|---|
| Oracle | 0.00% | 6.6m | 0.00% | 6.6m | 0.00% | 6.6m | 0.00% | 6.6m | 0.00% | 6.6m |
| URS | 24.05% | 7.5m | 8.11% | 5.3m | 20.96% | 8.1m | 48.02% | 7.8m | 17.35% | 5.8m |

| Method | AMDOCVRPTW Gap | AMDOCVRPTW Time | AMDOCVRPB Gap | AMDOCVRPB Time | AMDOCVRPBTW Gap | AMDOCVRPBTW Time | AMDCVRPBTW Gap | AMDCVRPBTW Time | AMDCVRPBP Gap | AMDCVRPBP Time |
|---|---|---|---|---|---|---|---|---|---|---|
| Oracle | 0.00% | 6.6m | 0.00% | 6.6m | 0.00% | 6.6m | 0.00% | 6.6m | 0.00% | 6.6m |
| URS | 26.02% | 7.3m | 27.78% | 6.5m | 23.67% | 6.2m | 17.52% | 6.8m | 36.18% | 6.5m |

| Method | AMDOCVRPBP Gap | AMDOCVRPBP Time | AMDOCVRPBPTW Gap | AMDOCVRPBPTW Time | AMDCVRPBPTW Gap | AMDCVRPBPTW Time | PDCVRP Gap | PDCVRP Time | OPDCVRP Gap | OPDCVRP Time |
|---|---|---|---|---|---|---|---|---|---|---|
| Oracle | 0.00% | 6.6m | 0.00% | 6.6m | 0.00% | 6.6m | 0.00% | 6.6m | 0.00% | 6.6m |
| URS | 39.31% | 6.7m | 24.99% | 6.9m | 20.98% | 7.6m | -1.47% | 4.3s | 4.93% | 4.3s |

| Method | APDCVRP Gap | APDCVRP Time | AOPDCVRP Gap | AOPDCVRP Time | AOP Gap | AOP Time | APCTSP Gap | APCTSP Time | ASPCTSP Value | ASPCTSP Time |
|---|---|---|---|---|---|---|---|---|---|---|
| Oracle | 0.00% | 6.6m | 0.00% | 6.6m | 0.00% | 6.6m | 0.00% | 6.6m | — | |
| URS | 7.03% | 1.2m | 11.13% | 1.2m | -5.47% | 1.2m | 43.79% | 1.2m | 2.22 | 1.2m |

# J. More Experiments

## J.1. Training Problem Selection

To empirically validate our selection strategy of training problems, we train two additional variants and compare them with our original URS: URS-Diff (Different Selection) and URS-Expand (Broader Problems). The first variant, URS-Diff, uses a different problem set where OCVRPTW is replaced by attribute-independent CVRPL. This modification means that the Open Route and Time Window attributes each appear in only one training problem. The second variant, URS-Expand, is trained on a broader set of 22 problems, incorporating all variants evaluated in MTPOMO (Liu et al., 2024) and MVMoE (Zhou et al., 2024).

As shown in Table 17, URS-Diff performs significantly worse on unseen problems involving Open Route and Time Window constraints. Meanwhile, URS-Expand achieves a slightly better performance than URS due to seeing more problem variants during training. However, the original URS remains highly competitive. Remarkably, despite being trained on only half the number of problem types (11 vs. 22), URS achieves the best solution on 12 out of the 22 evaluated problems.

These results demonstrate that the choice of our selected 11 problems offers a balanced trade-off, providing sufficient attribute coverage for strong generalization while maintaining training efficiency. Expanding the training set yields only marginal gains, while a poorly selected set leads to poor cross-problem generalization. In future work, we plan to explore more effective training problem selection strategies to further enhance model performance and training efficiency.

*Table 17.* Comparison under different training problem settings. The symbol * indicates the original problem selection, and $\Delta$ represents a different problem selection of equal number. All test problems are encompassed within URS-Expand.

| Method | ATSP (* $\Delta$) | TSP(* $\Delta$) | OP(* $\Delta$) | PCTSP(* $\Delta$) | PDTSP(* $\Delta$) | ACVRP(* $\Delta$) |
|---|---|---|---|---|---|---|
| URS-Diff | 4.52% | 1.05% | 0.82% | 1.31% | 6.09% | 5.10% |
| URS-Expand | 4.77% | 0.93% | 0.99% | 1.41% | 5.85% | 5.48% |
| URS | **2.26%** | **0.57%** | **0.45%** | **1.06%** | **4.98%** | **3.06%** |

| | CVRP(* $\Delta$) | CVRPTW(* $\Delta$) | CVRPB(* $\Delta$) | OCVRP(* $\Delta$) | OCVRPTW(*) | CVRPL($\Delta$) |
|---|---|---|---|---|---|---|
| URS-Diff | 1.97% | 6.26% | 1.80% | 3.75% | 15.45% | 0.58% |
| URS-Expand | 2.04% | **5.62%** | 1.77% | 3.45% | **4.34%** | 0.65% |
| URS | **1.81%** | 6.13% | **1.46%** | **3.24%** | 5.07% | **0.43%** |

| | OCVRPB | CVRPBL | CVRPLTW | OCVRPBTW | CVRPBLTW | OCVRPL |
|---|---|---|---|---|---|---|
| URS-Diff | 7.55% | 2.02% | 2.76% | 27.48% | 10.25% | 3.73% |
| URS-Expand | **5.39%** | 1.88% | **2.03%** | **9.13%** | **6.29%** | 3.41% |
| URS | 9.35% | **1.65%** | 2.67% | 13.77% | 9.18% | **3.22%** |

| | CVRPBTW | OCVRPBL | OCVRPLTW | OCVRPBLTW | Avg.gap | Best Sol. |
|---|---|---|---|---|---|---|
| URS-Diff | 10.00% | 7.56% | 15.61% | 27.80% | 7.43% | 0/22 |
| URS-Expand | **6.07%** | **5.47%** | **4.38%** | **9.23%** | **4.12%** | 10/22 |
| URS | 8.94% | 9.47% | 5.12% | 13.72% | 4.89% | **12/22** |

## J.2. Comparison with RouteFinder and CaDA

We strictly adhered to the problem settings established by MVMoE (Zhou et al., 2024) and ReLD (Huang et al., 2025) to ensure a fair and consistent evaluation. Notably, as shown in Table 6, URS already outperforms RF (Berto et al., 2025) and CaDA (Li et al., 2025a) on the large-scale CVRPLib benchmark dataset. In our cross-problem test, RF and CaDA are excluded due to discrepancies in the problems, including the number of trained CVRP variants (16 in their work vs. 5 in ours) and constraint settings (i.e., Backhauls and Time Windows). To ensure a fair comparison with RF and CaDA, we conduct two additional experiments:

- We evaluate URS against CaDA and the best-performing RouteFinder variant (RF-TE) on a shared subset of both seen and unseen problems, strictly restricting the evaluation to variants with perfectly aligned constraint definitions across all methods. As shown in Table 18, when evaluated under strictly matched conditions, URS still performs best overall among comparable methods. Note that URS can provide the strongest zero-shot generalization across all shared unseen problems.

- We retrain both RouteFinder-TE (denoted as RF-Retrain) and CaDA (denoted as CaDA-Retrain) from scratch using the exact same training settings as our URS and evaluate the three models on 48 CVRP variants. Since RF fails to train beyond 150 epochs due to the occurrence of NaN probabilities in our training setting, we evaluate all three methods at exactly 150 epochs to ensure a fair comparison. The experimental results in Table 19 show that under consistent training settings, URS still achieves the best overall performance among the three methods and provides the best solutions for the majority of problems (32/48), highlighting our strong zero-shot generalization.

*Table 18.* Comparison of shared seen and unseen VRP variants between URS, RF-TE, and CaDA.

| Method | Seen | | Unseen | | | | Avg.gap | Best Sol. |
|---|---|---|---|---|---|---|---|---|
| | CVRP | OCVRP | MDCVRP | MDOCVRP | MDCVRPL | MDOCVRPL | | |
| RF-TE | 2.06% | 2.86% | 63.60% | 33.52% | 66.75% | 34.40% | 33.86% | 0/6 |
| CaDA | 2.17% | **2.80%** | 40.96% | 53.21% | 42.40% | 53.34% | 32.48% | 1/6 |
| URS | **1.81%** | 3.24% | **15.04%** | **22.01%** | **15.32%** | **22.15%** | **13.26%** | **5/6** |

## J.3. Detailed Results on CVRPLib Set-X Dataset

We further evaluate the generalization ability of URS on benchmark instances from CVRPLib Set-X (Uchoa et al., 2017) with scale $\in [500, 1000]$. Our test results on CVRPLib Set-X datasets are presented in Table 20. All models are trained on instances of size $N = 100$. In the Set-X dataset, some results for comparative models are sourced from MVMoE (Zhou et al., 2024) and RouteFinder (Berto et al., 2025). As shown in Table 20, URS maintains its position as the best overall performance across instances of varying scales. Notably, RL-based URS surpasses the SL-based representative specialist model LEHD (8.678% vs. 12.836%) and many representative multi-task neural solvers (e.g., MVMoE and RouteFinder), underscoring its practical applicability in real-world scenarios.

*Table 19.* Comparison between URS and the retrained RouteFinder and CaDA on all 48 CVRP variants. Note that all models are trained for 150 epochs to ensure a fair comparison.

| Method | CVRPTW | | OCVRP | | CVRP | | CVRPB | | OCVRPTW | |
|---|---|---|---|---|---|---|---|---|---|---|
| | Gap | Time | Gap | Time | Gap | Time | Gap | Time | Gap | Time |
| RF-Retrain | 8.38% | 8s | 6.03% | 6s | 3.58% | 6s | 4.47% | 5s | 8.18% | 7s |
| CaDA-Retrain | **5.28%** | 9s | **3.22%** | 7s | **1.92%** | 7s | **1.61%** | 6s | **4.42%** | 8s |
| URS | 6.09% | 8s | 3.71% | 6s | 2.07% | 6s | 1.90% | 5s | 4.89% | 8s |

| Method | OCVRPB | | CVRPBL | | CVRPLTW | | OCVRPBTW | | CVRPBLTW | |
|---|---|---|---|---|---|---|---|---|---|---|
| | Gap | Time | Gap | Time | Gap | Time | Gap | Time | Gap | Time |
| RF-Retrain | 11.61% | 5s | 4.73% | 6s | 4.77% | 9s | 14.17% | 7s | 10.11% | 8s |
| CaDA-Retrain | 8.46% | 6s | 4.42% | 7s | 4.63% | 11s | **10.39%** | 8s | 10.87% | 10s |
| URS | **8.01%** | 6s | **2.12%** | 6s | **2.53%** | 9s | 12.37% | 7s | **8.74%** | 8s |

| Method | OCVRPL | | CVRPBTW | | OCVRPBL | | OCVRPLTW | | OCVRPBLTW | |
|---|---|---|---|---|---|---|---|---|---|---|
| | Gap | Time | Gap | Time | Gap | Time | Gap | Time | Gap | Time |
| RF-Retrain | 5.98% | 7s | 9.74% | 7s | 11.58% | 6s | 8.34% | 8s | 14.46% | 7s |
| CaDA-Retrain | 5.81% | 8s | **7.17%** | 8s | 12.48% | 7s | 8.55% | 11s | 15.94% | 10s |
| URS | **3.72%** | 7s | 8.42% | 7s | **8.05%** | 6s | **5.06%** | 8s | **12.37%** | 8s |

| Method | CVRPBP | | OCVRPBP | | CVRPBPL | | OCVRPBPTW | | CVRPBPLTW | |
|---|---|---|---|---|---|---|---|---|---|---|
| | Gap | Time | Gap | Time | Gap | Time | Gap | Time | Gap | Time |
| RF-Retrain | 14.40% | 7s | 18.31% | 7s | 14.01% | 7s | 11.43% | 9s | 18.70% | 10s |
| CaDA-Retrain | **13.37%** | 8s | 18.25% | 8s | 16.71% | 9s | **9.64%** | 10s | 20.17% | 12s |
| URS | 13.37% | 7s | **16.08%** | 7s | **12.94%** | 8s | 10.07% | 9s | **18.47%** | 10s |

| Method | CVRPBPTW | | OCVRPBPL | | OCVRPBPLTW | | CVRPL | | MDCVRP | |
|---|---|---|---|---|---|---|---|---|---|---|
| | Gap | Time | Gap | Time | Gap | Time | Gap | Time | Gap | Time |
| RF-Retrain | 18.84% | 9s | 17.76% | 8s | 10.88% | 9s | 2.21% | 7s | 27.10% | 18s |
| CaDA-Retrain | **17.11%** | 10s | 21.44% | 9s | 13.54% | 12s | 2.29% | 8s | **27.09%** | 23s |
| URS | 18.63% | 9s | **15.42%** | 8s | **9.57%** | 10s | **0.66%** | 7s | 38.17% | 23s |

| Method | MDCVRPTW | | MDOCVRP | | MDCVRPL | | MDCVRPB | | MDOCVRPTW | |
|---|---|---|---|---|---|---|---|---|---|---|
| | Gap | Time | Gap | Time | Gap | Time | Gap | Time | Gap | Time |
| RF-Retrain | 76.46% | 25s | **19.09%** | 18s | **27.41%** | 34s | 21.40% | 15s | 35.23% | 22s |
| CaDA-Retrain | 51.08% | 30s | 30.01% | 19s | 27.59% | 21s | **16.47%** | 19s | 54.46% | 29s |
| URS | **38.25%** | 31s | 21.34% | 21s | 39.22% | 26s | 20.83% | 18s | **28.40%** | 22s |

| Method | MDOCVRPB | | MDCVRPBL | | MDCVRPLTW | | MDOCVRPBTW | | MDCVRPBLTW | |
|---|---|---|---|---|---|---|---|---|---|---|
| | Gap | Time | Gap | Time | Gap | Time | Gap | Time | Gap | Time |
| RF-Retrain | 17.84% | 15s | 20.02% | 29s | 79.33% | 40s | 44.11% | 19s | 82.19% | 23s |
| CaDA-Retrain | 26.72% | 16s | **15.02%** | 18s | 43.43% | 29s | 64.29% | 25s | 50.11% | 24s |
| URS | **15.20%** | 17s | 17.36% | 21s | **39.98%** | 33s | **25.68%** | 19s | **33.24%** | 27s |

| Method | MDOCVRPL | | MDCVRPBTW | | MDOCVRPBL | | MDOCVRPLTW | | MDOCVRPBLTW | |
|---|---|---|---|---|---|---|---|---|---|---|
| | Gap | Time | Gap | Time | Gap | Time | Gap | Time | Gap | Time |
| RF-Retrain | **19.02%** | 19s | 84.94% | 21s | 17.72% | 16s | 35.06% | 24s | 44.12% | 21s |
| CaDA-Retrain | 30.61% | 20s | 61.39% | 25s | 26.94% | 17s | 39.49% | 29s | 49.12% | 25s |
| URS | 21.56% | 24s | **33.58%** | 25s | **14.84%** | 19s | **28.60%** | 24s | **25.84%** | 21s |

| Method | MDCVRPBP | | MDOCVRPBP | | MDCVRPBPL | | MDOCVRPBPTW | | MDCVRPBPLTW | |
|---|---|---|---|---|---|---|---|---|---|---|
| | Gap | Time | Gap | Time | Gap | Time | Gap | Time | Gap | Time |
| RF-Retrain | 54.74% | 21s | 36.08% | 21s | 54.19% | 23s | 47.24% | 26s | 84.51% | 30s |
| CaDA-Retrain | 54.45% | 25s | 52.38% | 22s | 50.69% | 24s | 69.07% | 32s | 55.63% | 32s |
| URS | **41.49%** | 24s | **24.18%** | 23s | **41.08%** | 27s | **24.19%** | 25s | **33.99%** | 35s |

| Method | MDCVRPBPTW | | MDOCVRPBPL | | MDOCVRPBPLTW | | Avg.gap | | Best Solutions | |
|---|---|---|---|---|---|---|---|---|---|---|
| | Gap | Time | Gap | Time | Gap | Time | | | | |
| RF-Retrain | 85.12% | 28s | 36.06% | 23s | 47.63% | 28s | 29.24% | | 3/48 | |
| CaDA-Retrain | 66.78% | 33s | 53.48% | 23s | 53.06% | 31s | 28.45% | | 13/48 | |
| URS | **34.55%** | 32s | **24.10%** | 25s | **24.32%** | 27s | **19.01%** | | **32/48** | |

*Table 20.* Results on large-scale CVRPLib instances (Set-X) (Uchoa et al., 2017) ($500 \leq N \leq 1000$). All models are trained on instances of size $N = 100$.

| Set-X Instance | Opt. | POMO Obj. | POMO Gap | LEHD Obj. | LEHD Gap | MTPOMO Obj. | MTPOMO Gap | MVMoE/4E Obj. | MVMoE/4E Gap | MVMoE/4E-L Obj. | MVMoE/4E-L Gap | RF-MVMoE Obj. | RF-MVMoE Gap | RF-TE Obj. | RF-TE Gap | URS Obj. | URS Gap |
|---|---|---|---|---|---|---|---|---|---|---|---|---|---|---|---|---|---|
| X-n502-k39 | 69226 | 75617 | 9.232% | 71438 | 3.195% | 77284 | 11.640% | 73533 | 6.222% | 74429 | 7.516% | 76338 | 10.274% | 71791 | 3.705% | **71281** | **2.969%** |
| X-n513-k21 | 24201 | 30518 | 26.102% | **25624** | **5.880%** | 28510 | 17.805% | 32102 | 32.647% | 31231 | 29.048% | 32639 | 34.866% | 28465 | 17.619% | 26166 | 8.119% |
| X-n524-k153 | 154593 | 201877 | 30.586% | 280556 | 81.480% | 192249 | 24.358% | 186540 | 20.665% | 182392 | 17.982% | **170999** | **10.612%** | 174381 | 12.800% | 175250 | 13.362% |
| X-n536-k96 | 94846 | 106073 | 11.837% | 103785 | 9.425% | 106514 | 12.302% | 109581 | 15.536% | 108543 | 14.441% | 105847 | 11.599% | 103272 | 8.884% | **102969** | **8.564%** |
| X-n548-k50 | 86700 | 103093 | 18.908% | 90644 | 4.549% | 94562 | 9.068% | 95894 | 10.604% | 95917 | 10.631% | 104289 | 20.287% | 100956 | 16.443% | **89768** | **3.539%** |
| X-n561-k42 | 42717 | 49370 | 15.575% | **44728** | **4.708%** | 47846 | 12.007% | 56008 | 31.114% | 51810 | 21.287% | 53383 | 24.969% | 49454 | 15.771% | 45964 | 7.601% |
| X-n573-k30 | 50673 | 83545 | 64.871% | **53482** | **5.543%** | 60913 | 20.208% | 59473 | 17.366% | 57042 | 12.569% | 61524 | 21.414% | 55952 | 10.418% | 54361 | 7.278% |
| X-n586-k159 | 190316 | 229887 | 20.792% | 232867 | 22.358% | 208893 | 9.761% | 215668 | 13.321% | 214577 | 12.748% | 212151 | 11.473% | 205575 | 8.018% | **202645** | **6.478%** |
| X-n599-k92 | 108451 | 150572 | 38.839% | 115377 | 6.386% | 120333 | 10.956% | 128949 | 18.901% | 125279 | 15.517% | 126578 | 16.714% | 116560 | 7.477% | **114423** | **5.507%** |
| X-n613-k62 | 59535 | 68451 | 14.976% | **62484** | **4.953%** | 67984 | 14.192% | 82586 | 38.718% | 74945 | 25.884% | 73456 | 23.383% | 67267 | 12.987% | 65901 | 10.693% |
| X-n627-k43 | 62164 | 84434 | 35.825% | 67568 | 8.693% | 73060 | 17.528% | 70987 | 14.193% | 70905 | 14.061% | 70414 | 13.271% | 67572 | 8.700% | **66499** | **6.973%** |
| X-n641-k35 | 63682 | 75573 | 18.672% | 68249 | 7.172% | 72643 | 14.071% | 75329 | 18.289% | 72655 | 14.090% | 71975 | 13.023% | 70831 | 11.226% | **67005** | **5.218%** |
| X-n655-k131 | 106780 | 127211 | 19.134% | 117532 | 10.069% | 116988 | 9.560% | 117678 | 10.206% | 118475 | 10.952% | 119057 | 11.497% | 112202 | 5.078% | **110237** | **3.237%** |
| X-n670-k130 | 146332 | 208079 | 42.197% | 220927 | 50.977% | 190118 | 29.922% | 197695 | 35.100% | 183447 | 25.364% | **168226** | **14.962%** | 168999 | 15.490% | 184010 | 25.748% |
| X-n685-k75 | 68205 | 79482 | 16.534% | **72946** | **6.951%** | 80892 | 18.601% | 97388 | 42.787% | 89441 | 31.136% | 82269 | 20.620% | 77847 | 14.137% | 75942 | 11.344% |
| X-n701-k44 | 81923 | 97843 | 19.433% | 86327 | 5.376% | 92075 | 12.392% | 98469 | 20.197% | 94924 | 15.870% | 90189 | 10.090% | 89932 | 9.776% | **86038** | **5.023%** |
| X-n716-k35 | 43373 | 51381 | 18.463% | 46502 | 7.214% | 52709 | 21.525% | 56773 | 30.895% | 52305 | 20.593% | 52250 | 20.467% | 49669 | 14.516% | **46496** | **7.200%** |
| X-n733-k159 | 136187 | 159098 | 16.823% | 149115 | 9.493% | 161961 | 18.925% | 178322 | 30.939% | 167477 | 22.976% | 156387 | 14.833% | 148463 | 9.014% | **147743** | **8.485%** |
| X-n749-k98 | 77269 | 87786 | 13.611% | **83439** | **7.985%** | 90582 | 17.229% | 100438 | 29.985% | 94497 | 22.296% | 92147 | 19.255% | 85171 | 10.227% | 83759 | 8.399% |
| X-n766-k71 | 114417 | 135464 | 18.395% | 131487 | 14.919% | 144041 | 25.891% | 152352 | 33.155% | 136255 | 19.086% | 130505 | 14.061% | **129935** | **13.563%** | 139371 | 21.810% |
| X-n783-k48 | 72386 | 90289 | 24.733% | **76766** | **6.051%** | 83169 | 14.897% | 100383 | 38.677% | 92960 | 28.423% | 96336 | 33.087% | 83185 | 14.919% | 77437 | 6.978% |
| X-n801-k40 | 73305 | 124278 | 69.536% | 77546 | 5.785% | 85077 | 16.059% | 91560 | 24.903% | 87662 | 19.585% | 87118 | 18.843% | 86164 | 17.542% | **77369** | **5.544%** |
| X-n819-k171 | 158121 | 193451 | 22.344% | 178558 | 12.925% | 177157 | 12.039% | 183599 | 16.113% | 185832 | 17.525% | 179596 | 13.581% | 174441 | 10.321% | **171024** | **8.160%** |
| X-n837-k142 | 193737 | 237884 | 22.787% | 207709 | 7.212% | 214207 | 10.566% | 229526 | 18.473% | 221286 | 14.220% | 230362 | 18.904% | 208528 | 7.635% | **203457** | **5.017%** |
| X-n856-k95 | 88965 | 152528 | 71.447% | **92936** | **4.464%** | 101774 | 14.398% | 99129 | 11.425% | 106816 | 20.065% | 105801 | 18.924% | 98291 | 10.483% | 94547 | 6.274% |
| X-n876-k59 | 99299 | 119764 | 20.609% | **104183** | **4.918%** | 116617 | 17.440% | 119619 | 20.463% | 114333 | 15.140% | 114016 | 14.821% | 107416 | 8.174% | 105417 | 6.161% |
| X-n895-k37 | 53860 | 70245 | 30.421% | **58028** | **7.739%** | 65587 | 21.773% | 79018 | 46.710% | 64310 | 19.402% | 69099 | 28.294% | 64871 | 20.444% | 58137 | 7.941% |
| X-n916-k207 | 329179 | 399372 | 21.324% | 385208 | 17.021% | 361719 | 9.885% | 383681 | 16.557% | 374016 | 13.621% | 373600 | 13.494% | 352998 | 7.236% | **346556** | **5.279%** |
| X-n936-k151 | 132715 | 237625 | 79.049% | 196547 | 48.097% | 186262 | 40.347% | 220926 | 66.466% | 190407 | 43.471% | **161343** | **21.571%** | 163162 | 22.942% | 172675 | 30.110% |
| X-n957-k87 | 85465 | 130850 | 53.104% | **90295** | **5.651%** | 98198 | 14.898% | 113882 | 33.250% | 105629 | 23.593% | 123633 | 44.659% | 102689 | 20.153% | 90485 | 5.874% |
| X-n979-k58 | 118976 | 147687 | 24.132% | 127972 | 7.561% | 138092 | 16.067% | 146347 | 23.005% | 139682 | 17.404% | 131754 | 10.740% | 129952 | 9.225% | **125353** | **5.360%** |
| X-n1001-k43 | 72355 | 100399 | 38.759% | **76689** | **5.990%** | 87660 | 21.153% | 114448 | 58.176% | 94734 | 30.929% | 88969 | 22.962% | 85929 | 18.760% | 77739 | 7.441% |
| Avg. Gap | | | 29.658% | | 12.836% | | 16.796% | | 26.408% | | 19.607% | | 18.795% | | 12.303% | | **8.678%** |

# K. Ablation Study

In this section, we conduct a detailed ablation study and analysis to demonstrate the effectiveness and robustness of URS. Please note that, unless stated otherwise, the results presented in the ablation study reflect the best result from multiple trajectories, and we employ instance augmentation to improve performance in the ablation study. We adopt the widely used 100-node setting as our primary evaluation scenario for URS.

## K.1. Effects of Node-type Indicator

*Table 21.* The ablation study of the node-type indicator on trained VRPs

| Method | ATSP | TSP | OP | PCTSP | PDTSP | ACVRP | CVRP | CVRPTW | CVRPB | OCVRP | OCVRPTW | Avg.gap |
|---|---|---|---|---|---|---|---|---|---|---|---|---|
| w/o $\xi_i$ | **2.14%** | **0.48%** | 0.45% | 1.06% | 9.28% | **2.84%** | 3.66% | 6.61% | 1.57% | 7.87% | 5.66% | 3.78% |
| w/ $\xi_i$ | 2.26% | 0.57% | **0.45%** | **1.06%** | 4.98% | 3.06% | **1.81%** | **6.13%** | **1.46%** | **3.24%** | **5.07%** | **2.74%** |

To evaluate the effectiveness of node-type indicator $\xi$, we conduct an ablation study about including and excluding $\xi$. Table 21 shows that adding the node-type indicator reduces the average optimality gap from 3.78% to 2.74%, with large gains on the routing problem with relatively rich instance node roles, such as PDTSP, where the coexistence of a depot and paired pickup-delivery nodes (with inherent precedence coupling) allows the explicit node-role encoding to further boost performance. Meanwhile, URS maintains a competitive performance in addressing the problem of single-node roles.

## K.2. Effects of Problem State

*Table 22.* Performance impact of removing the problem state feature $C_t$

| Method | ATSP | TSP | OP | PCTSP | PDTSP | ACVRP | CVRP | CVRPTW | CVRPB | OCVRP | OCVRPTW | Avg.gap |
|---|---|---|---|---|---|---|---|---|---|---|---|---|
| w/o $C_t$ | **2.09%** | **0.45%** | 0.60% | 1.20% | **4.59%** | 4.03% | 2.70% | **5.71%** | 2.64% | 4.02% | **4.62%** | 2.97% |
| w/ $C_t$ | 2.26% | 0.57% | **0.45%** | **1.06%** | 4.98% | **3.06%** | **1.81%** | 6.13% | **1.46%** | **3.24%** | 5.07% | **2.74%** |

As described in Appendix E.3, the majority of VRP variants are already solved without supplying an additional problem state $C_t$. To test whether the remaining use of $C_t$ matters, we conduct an ablation in which we eliminate it entirely, enforcing $C_t \equiv 0$ for all problems. The results in Table 22 show that URS maintains its competitive performance even in the absence of $C_t$. This indicates that explicit constraint-state features are not essential to URS effectiveness. Interestingly, on (O)CVRPTW instances, the model without any explicit constraint features outperforms the version that supplies only capacity information. We guess that capacity and time windows information are equally important, so enforcing only capacity creates a bias toward load considerations.

## K.3. Effects of Mixed Bias Module

*Table 23.* Ablation of prior components in the MBM.

| $D$ | $D^{\mathbf{T}}$ | $R$ | ATSP | TSP | OP | PCTSP | PDTSP | ACVRP | CVRP | CVRPTW | CVRPB | OCVRP | OCVRPTW | Avg.gap |
|---|---|---|---|---|---|---|---|---|---|---|---|---|---|---|
| ✓ | × | × | 16.64% | **0.44%** | **0.25%** | **0.67%** | 7.33% | 16.62% | 2.42% | 6.44% | 3.96% | 9.44% | 5.47% | 6.33% |
| ✓ | ✓ | × | **2.09%** | 0.45% | 0.48% | 0.82% | 7.66% | 9.11% | 2.52% | 6.89% | 2.68% | 5.70% | 6.36% | 4.07% |
| ✓ | × | ✓ | 16.60% | 0.46% | 0.49% | 0.94% | **4.49%** | 26.22% | 4.16% | 6.57% | 3.07% | 12.23% | 5.53% | 7.34% |
| ✓ | ✓ | ✓ | 2.26% | 0.57% | 0.45% | 1.06% | 4.98% | **3.06%** | **1.81%** | **6.13%** | **1.46%** | **3.24%** | **5.07%** | **2.74%** |

To further assess the effectiveness of MBM, we conduct an ablation study that examines different combinations of problem-specific priors within MBM. Owing to its flexibility, the MBM allows multiple priors to be imposed on the same shared attention layer. The results in Table 23 underscore the complementary roles of the three prior components: (1) The transposed distance matrix $D^{\mathbf{T}}$ captures directional asymmetry absent from the raw distance matrix, and its inclusion markedly stabilizes optimization on asymmetric variants; (2) Removing the pickup-delivery relation matrix $R$ degrades performance on PDTSPs, confirming that explicit relational coupling is indispensable for these instances; (3) When all priors are fused, MBM can simultaneously learn the geometric and relational biases inherent in various problems, yielding the best overall performance.

## K.4. Effects of Different Decoder Architectures

*Table 24.* Effect of different decoder architectures on URS cross-problem performance.

| Decoder | ATSP | TSP | OP | PCTSP | PDTSP | ACVRP | CVRP | CVRPTW | CVRPB | OCVRP | OCVRPTW | Avg.gap |
|---|---|---|---|---|---|---|---|---|---|---|---|---|
| URS-POMO | 4.83% | 0.74% | 0.72% | 1.29% | 5.45% | 5.09% | 1.99% | 6.42% | 1.83% | 3.62% | 5.31% | 3.39% |
| URS-ReLD | 6.48% | 0.90% | 0.86% | 1.51% | 5.81% | 6.51% | 2.33% | 6.56% | 2.33% | 3.62% | 5.64% | 3.87% |
| URS | **2.26%** | **0.57%** | **0.45%** | **1.06%** | **4.98%** | **3.06%** | **1.81%** | **6.13%** | **1.46%** | **3.24%** | **5.07%** | **2.74%** |

We adopt an AM (Kool et al., 2019) as our basic encoder-decoder model. Multiple decoder refinements have been proposed around AM (Kwon et al., 2020; Huang et al., 2025; Zhou et al., 2026). To assess how decoder structure impacts multi-task performance, we additionally train two URS variants whose decoders follow POMO and ReLD, denoted URS-POMO and URS-ReLD. For stronger and fairer baselines, we apply Distance-aware Attention Reshaping (DAR) (Wang et al., 2025b) to both, and all decoder parameters are generated by a hypernetwork conditioned on the problem representation $\lambda$ (see Appendix E.2). As shown in Table 24, URS-POMO and URS-ReLD still exhibit strong cross-problem robustness, while replacing the decoder with ICAM (Zhou et al., 2026) yields a further overall improvement.

Furthermore, the results in Table 24 reveal a notable optimization challenge. By incorporating an additional Feed-Forward Network layer compared to POMO, the ReLD decoder forces the hypernetwork to map to a larger parameter space, thereby increasing its learning difficulty and degrading performance. This optimization difficulty prevents URS-ReLD from matching the performance of the original ReLD-MTL, where weights are learned directly rather than being dynamically generated. Given the outstanding performance of ReLD and other heavy-decoder paradigms in single-task optimization (Luo et al., 2023; Drakulic et al., 2023), exploring how to adapt problem-conditioned generation mechanisms for heavier decoders is a highly promising direction.

## K.5. Effects of MBM & Parameter Generator on Zero-shot Generalization

*Table 25.* Ablation of MBM, WEIGHT($\lambda$), and BIAS($\lambda$) on cross-problem zero-shot generalization. The symbol (*) indicates the seen problems in training.

| MBM | WEIGHT($\lambda$) | BIAS($\lambda$) | ATSP (*) | TSP (*) | OP (*) | PCTSP (*) | PDTSP (*) | ACVRP (*) |
|---|---|---|---|---|---|---|---|---|
| × | ✓ | ✓ | 16.64% | **0.44%** | **0.25%** | **0.67%** | 7.33% | 16.62% |
| ✓ | × | × | 4.24% | 1.10% | 1.04% | 1.42% | 6.38% | 4.07% |
| ✓ | × | ✓ | 3.83% | 1.16% | 1.09% | 1.52% | 6.65% | 4.27% |
| ✓ | ✓ | × | 2.79% | 1.58% | 1.39% | 1.92% | 7.05% | 3.39% |
| ✓ | ✓ | ✓ | **2.26%** | 0.57% | 0.45% | 1.06% | **4.98%** | **3.06%** |

| MBM | WEIGHT($\lambda$) | BIAS($\lambda$) | CVRP (*) | CVRPTW (*) | CVRPB (*) | OCVRP (*) | OCVRPTW (*) | CVRPL |
|---|---|---|---|---|---|---|---|---|
| × | ✓ | ✓ | 2.42% | 6.44% | 3.96% | 9.44% | 5.47% | 0.98% |
| ✓ | × | × | 6.23% | 7.22% | 5.00% | 3.99% | 5.41% | 2.92% |
| ✓ | × | ✓ | 4.94% | **6.08%** | 3.00% | 4.61% | 5.21% | 4.79% |
| ✓ | ✓ | × | 2.90% | 6.60% | 3.12% | 4.47% | **4.76%** | 1.46% |
| ✓ | ✓ | ✓ | **1.81%** | 6.13% | **1.46%** | **3.24%** | 5.07% | **0.43%** |

| MBM | WEIGHT($\lambda$) | BIAS($\lambda$) | OCVRPB | CVRPBL | CVRPLTW | OCVRPBTW | CVRPBLTW | OCVRPL |
|---|---|---|---|---|---|---|---|---|
| × | ✓ | ✓ | 15.95% | 4.18% | 2.89% | 18.78% | 10.93% | 9.51% |
| ✓ | × | × | 9.54% | 5.30% | 2.73% | 14.37% | 9.80% | 4.64% |
| ✓ | × | ✓ | 9.58% | 3.25% | 3.07% | 14.16% | 9.91% | 3.88% |
| ✓ | ✓ | × | 9.92% | 3.34% | 3.08% | 14.02% | 9.92% | 4.52% |
| ✓ | ✓ | ✓ | **9.35%** | **1.65%** | **2.67%** | **13.77%** | **9.18%** | **3.22%** |

| MBM | WEIGHT($\lambda$) | BIAS($\lambda$) | CVRPBTW | OCVRPBL | OCVRPLTW | OCVRPBLTW | Avg.gap | Best Sol. |
|---|---|---|---|---|---|---|---|---|
| × | ✓ | ✓ | 10.74% | 16.14% | 5.57% | 18.80% | 8.37% | 3/22 |
| ✓ | × | × | 9.67% | 10.67% | 5.23% | 14.49% | 6.16% | 0/22 |
| ✓ | × | ✓ | 9.63% | 10.42% | 5.51% | 13.99% | 5.93% | 1/22 |
| ✓ | ✓ | × | 9.79% | 10.02% | 5.81% | 13.98% | 5.72% | 1/22 |
| ✓ | ✓ | ✓ | **8.94%** | **9.47%** | **5.12%** | **13.72%** | **4.89%** | **17/22** |

To empirically evaluate the necessity of these components, we have conducted tests on the 11 seen problems as well as 11

additional unseen CVRP variants widely investigated in current multi-task research (e.g., MVMoE (Zhou et al., 2024)). The results are presented in Table 25. We can observe that removing MBM leads to a significant performance drop on problems with complex geometric properties, such as ATSP, ACVRP, and PDTSP, which confirms MBM is critical for perceiving specific geometric biases. In addition, removing WEIGHT($\lambda$) or BIAS($\lambda$) also results in performance degradation across most tasks. In terms of overall performance, URS significantly outperforms any model where a single module is removed. The full URS model achieves the lowest average gap and provides the best solution on the majority of tasks (17/22). This demonstrates that the synergy of these components is essential for robust cross-problem generalization.

### K.6. Effects of Each UDR Component

To clarify the independent contribution of each UDR component, we conduct ablations by removing each component in turn. As shown in Table 26, the absence of any single component leads to notable performance degradation. A complete UDR achieves the lowest average gap and provides the best solution on the majority of tasks (15/22), affirming the necessity of our UDR design for robust zero-shot generalization.

*Table 26.* The ablation study of each UDR component. The symbol (*) indicates the seen problems in training. $\rho$, $\omega$, and $\xi$ represent positional identifier, unified attribute set, and node-type indicator, respectively. Since removing positional identifiers prevents the use of instance augmentation, we have excluded it from all results to ensure consistency in the inference strategy.

| $\rho$ | $\omega$ | $\xi$ | ATSP (*) | TSP (*) | OP (*) | PCTSP (*) | PDTSP (*) | ACVRP (*) |
|---|---|---|---|---|---|---|---|---|
| $\times$ | $\checkmark$ | $\checkmark$ | 7.64% | 1.76% | 1.17% | 1.96% | 8.54% | 7.16% |
| $\checkmark$ | $\times$ | $\checkmark$ | 10.07% | 1.98% | 2.70% | 7.05% | 13.50% | 13.36% |
| $\checkmark$ | $\checkmark$ | $\times$ | **5.43%** | **1.02%** | 1.08% | 1.86% | 12.72% | **5.81%** |
| $\checkmark$ | $\checkmark$ | $\checkmark$ | 5.51% | 1.17% | **0.96%** | **1.79%** | **7.71%** | 6.11% |

| $\rho$ | $\omega$ | $\xi$ | CVRP (*) | CVRPTW (*) | CVRPB (*) | OCVRP (*) | OCVRPTW (*) | CVRPL |
|---|---|---|---|---|---|---|---|---|
| $\times$ | $\checkmark$ | $\checkmark$ | 2.68% | 6.88% | 2.84% | 4.64% | **5.91%** | 1.29% |
| $\checkmark$ | $\times$ | $\checkmark$ | 4.15% | 23.71% | 5.24% | 7.10% | 19.10% | 2.70% |
| $\checkmark$ | $\checkmark$ | $\times$ | 4.72% | 7.67% | 2.83% | 9.64% | 6.96% | 3.33% |
| $\checkmark$ | $\checkmark$ | $\checkmark$ | **2.62%** | **6.80%** | **2.69%** | **4.54%** | 6.00% | **1.23%** |

| $\rho$ | $\omega$ | $\xi$ | OCVRPB | CVRPBL | CVRPLTW | OCVRPBTW | CVRPBLTW | OCVRPL |
|---|---|---|---|---|---|---|---|---|
| $\times$ | $\checkmark$ | $\checkmark$ | 11.87% | 3.12% | **3.34%** | 34.34% | 44.03% | 4.63% |
| $\checkmark$ | $\times$ | $\checkmark$ | 15.02% | 5.47% | 19.53% | 30.88% | 29.64% | 6.98% |
| $\checkmark$ | $\checkmark$ | $\times$ | 26.06% | **2.69%** | 4.13% | 34.62% | 26.55% | 9.68% |
| $\checkmark$ | $\checkmark$ | $\checkmark$ | **11.54%** | 3.03% | 3.38% | **15.47%** | **10.70%** | **4.55%** |

| $\rho$ | $\omega$ | $\xi$ | CVRPBTW | OCVRPBL | OCVRPLTW | OCVRPBLTW | Avg.gap | Best Sol. |
|---|---|---|---|---|---|---|---|---|
| $\times$ | $\checkmark$ | $\checkmark$ | 43.99% | 11.99% | **6.02%** | 34.66% | 11.38% | 3/22 |
| $\checkmark$ | $\times$ | $\checkmark$ | 29.34% | 14.84% | 19.22% | 31.02% | 14.21% | 0/22 |
| $\checkmark$ | $\checkmark$ | $\times$ | 28.60% | 25.96% | 7.00% | 35.93% | 12.01% | 4/22 |
| $\checkmark$ | $\checkmark$ | $\checkmark$ | **10.51%** | **11.68%** | 6.08% | **15.38%** | **6.34%** | **15/22** |

### K.7. Hypernetwork vs. Adapter

We train URS-Adapter, a URS variant that adopts a distinct decoder for each of the 11 variants, and compare it with the original URS using a hypernetwork on 11 seen problems (see Table 27) and large-scale CVRP variants (see Table 28). URS consistently outperforms URS-Adapter across 11 seen problems, achieving higher parameter efficiency. This performance advantage holds consistently on large-scale CVRP variants, which further solidifies our conclusion.

## L. Licenses for Used Resources

We list the existing codes and datasets in Table 29, all of which are open-source resources for academic use.

*Table 27.* Performance comparison between URS-Hypernet and URS-Adapter on 11 seen VRP variants.

| Method | OP | | PCTSP | | PDTSP | | ACVRP | |
|---|---|---|---|---|---|---|---|---|
| | Gap | Time | Gap | Time | Gap | Time | Gap | Time |
| URS-Adapter | 0.88% | 4s | 1.47% | 4s | 5.69% | 4s | 3.83% | 1.5m |
| URS | **0.45%** | 4s | **1.06%** | 4s | **4.98%** | 4s | **3.06%** | 1.5m |

| Method | CVRPTW | | CVRPB | | OCVRP | | OCVRPTW | |
|---|---|---|---|---|---|---|---|---|
| | Gap | Time | Gap | Time | Gap | Time | Gap | Time |
| URS-Adapter | 6.34% | 8s | 3.21% | 5s | 4.06% | 6s | **4.94%** | 7s |
| URS | **6.13%** | 8s | **1.46%** | 5s | **3.24%** | 6s | 5.07% | 7s |

| Method | ATSP | | TSP | | CVRP | | Average | Best |
|---|---|---|---|---|---|---|---|---|
| | Gap | Time | Gap | Time | Gap | Time | Gap | Solution |
| URS-Adapter | 2.97% | 1.2m | 0.82% | 4s | 2.85% | 6s | 3.37% | 1/11 |
| URS | **2.26%** | 1.2m | **0.57%** | 4s | **1.81%** | 6s | **2.74%** | **10/11** |

*Table 28.* Generalization comparison between URS and URS-Adapter across diverse large-scale CVRP variants.

| Problem | Method | N = 1000 | | N = 2000 | | N = 3000 | | N = 4000 | | N = 5000 | |
|---|---|---|---|---|---|---|---|---|---|---|---|
| | | Obj.(Gap) | Time | Obj.(Gap) | Time | Obj.(Gap) | Time | Obj.(Gap) | Time | Obj.(Gap) | Time |
| CVRPTW | PyVRP | 159.35(0.00%) | 20m | 337.15(0.00%) | 40m | 516.97(0.00%) | 1h | 618.58(0.00%) | 2.7h | 787.25(0.00%) | 3.3h |
| | URS-Adapter | 173.68(8.99%) | 1.3m | 367.22(8.92%) | 10.8m | 564.55(9.21%) | 30.8m | 677.91(9.59%) | 1.2h | 860.63(9.32%) | 2.3h |
| | URS | **172.58(8.30%)** | 1.3m | **365.27(8.34%)** | 11.1m | **559.03(8.14%)** | 30.1m | **673.63(8.90%)** | 1.2h | **853.79(8.45%)** | 2.3h |
| CVRPB | PyVRP | 36.52(0.00%) | 20m | 51.80(0.00%) | 40m | 75.12(0.00%) | 1h | 93.52(0.00%) | 2.7h | 112.21(0.00%) | 3.3h |
| | URS-Adapter | 42.71(16.96%) | 43s | 62.73(21.11%) | 5.5m | 85.72(14.11%) | 17.5m | 107.17(14.59%) | 41.9m | 127.93(14.01%) | 1.3h |
| | URS | **34.86(-4.54%)** | 40s | **50.39(-2.72%)** | 5.1m | **72.01(-4.15%)** | 17.3m | **91.74(-1.91%)** | 41.2m | **111.93(-0.24%)** | 1.3h |
| OCVRPTW | PyVRP | 90.91(0.00%) | 20m | 164.00(0.00%) | 40m | 224.16(0.00%) | 1h | 299.45(0.00%) | 2.7h | 367.86(0.00%) | 3.3h |
| | URS-Adapter | 105.96(16.56%) | 1.1m | 203.52(24.10%) | 9.9m | 287.70(28.35%) | 28.6m | 391.06(30.59%) | 1h | 483.19(31.35%) | 1.9h |
| | URS | **98.76(8.63%)** | 1m | **178.07(8.58%)** | 7.6m | **243.88(8.80%)** | 25.4m | **325.64(8.74%)** | 1h | **398.72(8.39%)** | 1.9h |

*Table 29.* List of licenses for the codes and datasets we used in this work.

| Resource | Type | Link | License |
|---|---|---|---|
| PyVRP (Wouda et al., 2024) | Code | https://github.com/PyVRP/PyVRP | MIT License |
| LKH3 (Helsgaun, 2017) | Code | http://webhotel4.ruc.dk/~keld/research/LKH-3/ | Available for academic research use |
| OR-Tools (Perron & Furnon, 2023) | Code | https://github.com/google/or-tools | Apache-2.0 License |
| POMO (Kwon et al., 2020) | Code | https://github.com/yd-kwon/POMO/tree/master/NEW_py_ver | MIT License |
| Sym-NCO (Kim et al., 2022) | Code | https://github.com/alstn12088/Sym-NCO | Available for any non-commercial use |
| LEHD (Luo et al., 2023) | Code | https://github.com/CIAM-Group/NCO_code/tree/main/single_objective/LEHD | Available for any non-commercial use |
| BQ (Drakulic et al., 2023) | Code | https://github.com/naver/bq-nco | CC BY-NC-SA 4.0 license |
| ICAM (Zhou et al., 2026) | Code | https://github.com/CIAM-Group/ICAM | MIT License |
| ELG (Gao et al., 2024) | Code | https://github.com/gaocrr/ELG | MIT License |
| MatNet (Kwon et al., 2021) | Code | https://github.com/yd-kwon/MatNet/tree/main | MIT License |
| Heter-AM (Li et al., 2021) | Code | https://github.com/jingwenli0312/Heterogeneous-Attentions-PDP-DRL | MIT License |
| MTPOMO (Liu et al., 2024) | Code | https://github.com/FeiLiu36/MTNCO | MIT License |
| MVMoE (Zhou et al., 2024) | Code | https://github.com/RoyalSkye/Routing-MVMoE | MIT License |
| RouteFinder (Berto et al., 2025) | Code | https://github.com/ai4co/routefinder | MIT License |
| CaDA (Li et al., 2025a) | Code | https://github.com/CIAM-Group/CaDA | MIT License |
| ReLD (Huang et al., 2025) | Code | https://github.com/ziweileonhuang/reld-nco | MIT License |
| GOAL (Drakulic et al., 2025) | Code | https://github.com/naver/goal-co | Available for any non-commercial use |
| CVRPLib Set-X (Uchoa et al., 2017) | Dataset | http://vrp.galgos.inf.puc-rio.br/index.php/en/ | Available for academic research use |
| CVRPLib Set-AGS (Arnold et al., 2019) | Dataset | http://vrp.galgos.inf.puc-rio.br/index.php/en/ | Available for academic research use |

