# OpenReview forum: "URS: A Unified Neural Routing Solver for Cross-Problem Zero-Shot Generalization"
_ICML.cc/2026/Conference — ICML 2026 regular_

### Official Review · Reviewer_kR1Y · 2026-02-25

**Soundness:** 2
**Presentation:** 3
**Significance:** 2
**Originality:** 2
**Overall Recommendation:** 3
**Confidence:** 4

**Summary:**

This paper proposes URS (Unified Neural Routing Solver), a reinforcement learning-based neural solver designed to address the cross-problem zero-shot generalization challenge in vehicle routing problems (VRPs). The authors introduce three key innovations: (1) a Unified Data Representation (UDR) that replaces explicit problem enumeration with data unification, (2) a Mixed Bias Module (MBM) to capture multiple geometric priors across different VRP variants, and (3) a problem-conditioned parameter generator for adaptive decoding. The method is evaluated on 110 VRP variants (including 99 unseen variants) and demonstrates scalability to large-scale instances with up to 7000 nodes. The authors claim URS is the first neural solver to handle over 100 VRP variants with a single model without retraining or fine-tuning.

**Compliance With Llm Reviewing Policy:**

Affirmed.

**Key Questions For Authors:**

1. Beyond empirical validation in Appendix I.1, is there a theoretical framework or principled approach for selecting the minimal set of training problems needed for effective zero-shot generalization?
2. For applications where peak performance on a specific VRP variant is critical, how should practitioners decide between using URS versus training a specialist solver? What is the performance gap threshold that would justify the additional training cost?
3. In which types of VRP variants or constraint combinations does URS struggle most? Are there specific problem characteristics that predict poor zero-shot generalization?
4. What is the theoretical computational complexity of the MBM compared to standard attention? How does inference time scale with the number of active constraints in the UDR?
5. Have the authors tested URS on real-world VRP instances (beyond synthetic benchmarks)? How does performance translate to practical logistics scenarios with noisy or incomplete data?

**Limitations:**

Yes

**Strengths And Weaknesses:**

Strengths:
1. Evaluation on different VRP variants exceeds prior work (typically 10-20 variants), providing strong evidence of broad generalization capability.
2. The ability to generalize to 99 unseen variants without fine-tuning represents a meaningful advance in multi-task neural combinatorial optimization.

Weaknesses:
1. On several seen problems (e.g., ATSP, PDTSP), URS underperforms compared to specialist solvers (Table 3), raising questions about the trade-off between generality and peak performance.
2. While Appendix I.1 discusses training problem selection, the rationale for choosing exactly 11 specific problems lacks theoretical grounding. The ablation shows performance is sensitive to this choice.
3. Different oracle solvers are used for different problems (LKH-3, PyVRP, OR-Tools, Compass, ILS), which may introduce variability in optimality gap calculations.

---

> ### Author Rebuttal · Authors · 2026-03-31
>
> Thank you very much for your time and effort in reviewing our work. We provide an anonymous link (https://anonymous.4open.science/r/ICML26_Submission26157/Reviewer_kR1Y.pdf) to present supplementary results.
>
> **W1/Q2: MTL vs. STL:** Thank you for pointing this out. In real-world applications, constraints are open-ended, and retraining a specialist solver for each new scenario incurs high training costs and is impractical. While URS underperforms URS-STL at the training scale (N=100), by learning the shared knowledge and structural similarities across different VRPs, multi-task learning can efficiently avoid overfitting to a specific scale of a specific problem and enable URS to achieve better scalability in more practical, large-scale scenarios (see **Table 1**).
>
> For our URS, if practitioners target a specific variant whose scale matches the training scale exactly, employing an STL model is preferable to obtain peak in-domain performance. However, if they are facing large-scale instances or unseen VRPs, URS-MTL may be a better choice.
>
> **W2/Q2: Training Problems Selection:** Thank you for pointing this out. We adhere to the principle of minimal redundancy and strictly align with existing work on CVRP variant selection (e.g., MVMoE and ReLD): most features in UDR appear in at least two training problems, helping prevent URS from overfitting to a single problem. The only exception is the Penalty attribute, as there is structural similarity between the seen PCTSP and unseen SPCTSP. We provide **Table 2** to detail the occurrence frequency of each feature in training.
>
> To verify the principle, we have conducted a new ablation study by removing OCVRP in training, meaning the "node in open route" indicator appears only once in OCVRPTW (see **Table 3**). This aligns with our findings in Appendix I.1: The selected 11 problems provide a balanced trade-off, providing sufficient attribute coverage for strong generalization while maintaining training efficiency.
>
> We totally agree with you that a rigorous theoretical analysis would be valuable in supporting our problem selection. However, this is currently lacking in related works. We leave it as an important topic for future work. In addition, we hope the above empirical analysis provides some insights into the problem selection.
>
> **W3: Oracle Solver:**  Thank you for pointing this out. Given the 110 VRP variants evaluated in this work, there is currently no universal heuristic that can address all of them. As shown in **Tables 10 and 11 in our manuscript**, for the Oralce selection, we follow the established works (e.g., MVMoE and RouteFinder) in widely studied symmetric VRPs. For less-explored asymmetric VRPs, we use OR-Tools as the unified oracle across all these variants to mitigate variability in gap calculations. We will explicitly state this rationale in the revised paper.
>
> **Q3: Generalization Boundary:** Thank you for pointing this out. Based on our results, URS struggles most with Duration-Limit (L) and Open-Route (O) constraints. We attribute this to the input attributes in UDR for CVRP, OCVRP, and CVRPL, which are largely identical, resulting in URS receiving similar input signals for tasks with diametrically opposed solution structures: OCVRP expands the search space of nodes, while CVRPL restricts it. We are aware of this potential ambiguity and introduce the node-type indicator by flagging "open route" nodes. Empirical results show that this is insufficient to fully resolve the ambiguity, and L-variants (which lack a dedicated flag) suffer even more from it. Developing a more expressive UDR to capture the problem disparities is a critical direction for our future work.
>
> **Q4.1: MBM vs. MHA:** Thank you for raising this concern. For a detailed comparison between MBM and existing attention mechanisms (e.g., MHA), please refer to the **response to Q3 of Reviewer VZ32**.
>
> **Q4.2 Inference Time:** Thank you for raising this concern. To quantify how inference time scales with the number of active constraints, we have conducted a progressive constraint-stacking experiment: Starting with the standard CVRP as the baseline, we incrementally added constraints, culminating in the complex AMDOCVRPBPLTW variant (**Figure 1**).
>
>
> **Q5: Real-world VRP instances:** Thank you for raising this concern. We have conducted a new empirical study using real-world CVRP instances derived from OpenStreetMap and provided by [1] across eight distinct cities (**Figure 2 and Table 4**). Additionally, URS performs well on SPCTSP, where the true prize is revealed only upon visitation.
>
> While URS successfully handles attribute-level uncertainty, we acknowledge that solving node-level uncertainty requires customized network design. We leave it as a promising direction for future work.
>
> We will add all of the above results and analyses in the revised paper.
>
> **References:**
>
> [1] Towards Real-World Routing with Neural Combinatorial Optimization, ICLR2026.

---

### Official Review · Reviewer_STCX · 2026-03-04

**Soundness:** 4
**Presentation:** 4
**Significance:** 3
**Originality:** 3
**Overall Recommendation:** 5
**Confidence:** 4

**Summary:**

This paper presents URS, a neural solver for zero-shot generalization to a large number of unseen VRPs. It uses a Unified Data Representation (UDR) to cover a wide range of node attributes, supporting encoding more than hundred VRP variants. It use a multi-hot problem code to condition the decoder via hypernetwork-generated weights and bias terms, and introduces a Mixed Bias Module (MBM) to inject structural priors (3 node-wise matrices) into attention. Experiments across many variants show better performance on seen variants and strong zero-shot performance on unseen problem variants.

**Compliance With Llm Reviewing Policy:**

Affirmed.

**Final Justification:**

I am happy to retain my positive score of 5.

**Key Questions For Authors:**

1. How sensitive is the hypernetwork to the input problem multi-hot code? This can be measured by the solver’s performance degradation when the multi-hot code is intentionally perturbed or mismatched relative to the problem being solved.

2. Does the hypernetwork sacrifice performance in single-task learning (STL)? This can be examined by comparing URS-STL with and without the hypernetwork

**Limitations:**

Yes.

**Strengths And Weaknesses:**

trengths:
* The method is well-motivated and internally consistent, with a coherent overall design.
* The experimental protocol is comprehensive and fair, and the results largely support the paper’s claims.
* The proposed UDR substantially expands the encodable variant space, and MBM effectively injects variant-specific structural priors into the decision process, greatly broadening the solvable problem scope (from dozens to hundreds).
* The problem-conditioned parameter generation (via a hypernetwork driven by a multi-hot problem code) reduces the extra cost of per-task adapters in existing methods.

Weaknesses:
* Although UDR and the multi-hot code enable many attributes (combinations), the supported attribute set is still fixed. Thus, the approach remains dependent on attribute/problem enumeration and cannot zero-shot generalize to new attributes/constraints outside this set. Nonetheless, it is still a significant step forward in scaling the number of variants handled by one solver.
* Despite strong zero-shot performance, URS appears consistently weaker on seen tasks than ReLD-MTL and MVMoE when their results are available. Since the paper targets multi-task learning, a deeper discussion of URS’s multi-task performance on seen tasks (beyond emphasizing the promising performance of the specialist solver URS-STL) would further strengthen the quality.
* From Tables 2, 3, 20, ReLD-MTL performs strongly on seen tasks, while combining ReLD with URS (URS-ReLD) can underperform ReLD-MTL. This may suggest URS needs specific adaptations when used on heavier decoders. Further discussion on this point is worthwhile.
* Other issues:
  * in Fig. 1, the arrows from BIAS(λ) and WEIGHT(λ) point to the whole encoder/decoder although they operate on specific components. Pointing to the exact locations would improve clarity.
  * Several tables contain "–" entries without explanation. Adding a caption note would help interpretation.

---

> ### Author Rebuttal · Authors · 2026-03-31
>
> Thank you very much for your time and effort in reviewing our work. We provide an anonymous link (https://anonymous.4open.science/r/ICML26_Submission26157/Reviewer_STCX.pdf) to present supplementary results.
>
> **W1: Generalize to New Attributes:** We sincerely appreciate your recognition of the value and scalability of our URS framework. We fully agree with your insightful observation. Generalizing to new attributes outside the current UDR space remains an open challenge, and bridging this gap to enable dynamic adaptation to completely out-of-distribution attributes is a critical next step for the NCO community. We regard it as one of our important future works.
>
> **W2: Detailed Analysis for Seen Problems:** Thank you for your insightful feedback. Since the primary objective is to improve zero-shot cross-problem generalization, we deliberately reduce the model's reliance on specific constraint states. Unlike ReLD-MTL and MVMoE, which explicitly embed multiple constraint states (i.e., C, TW, O, and L) into the decoder, URS introduces only one fundamental problem state for CVRP variants (i.e., remaining load) and delegates diverse constraints to a model-agnostic masking function. Due to the lack of constraint state awareness, this results in our performance on small-scale seen tasks being consistent with the training scale being inferior to that of ReLD-MTL and MVMoE. Notably, URS demonstrates a clear advantage in the large-scale seen problems and unseen constraints. We will revise our experimental discussion in the revised paper and focus on multi-task performance.
>
> **W3: URS-ReLD:** Thank you for pointing this out. In the URS framework, decoder parameters are dynamically generated by a hypernetwork. Because the ReLD decoder introduces an additional Feed-Forward Network layer compared to POMO, the hypernetwork must learn a larger parameter space, which increases its learning difficulty and degrades performance. This optimization difficulty prevents URS-ReLD from matching the performance of ReLD-MTL, where weights are learned directly rather than being dynamically generated.
>
> Given the outstanding performance of ReLD and other heavy-decoder paradigms in single-task optimization, exploring how to adapt problem-conditioned generation mechanisms for heavier decoders is a highly promising direction for our future work. We will include the discussion in the revised manuscript.
>
> **W4: Other Issues:** We truly appreciate your constructive suggestions, and we believe this correction significantly improves the quality of our paper. We have carefully incorporated your suggestions and revised our manuscript. Additionally, to prevent any reader confusion, we will carefully review the manuscript and revise any inappropriate or ambiguous expressions.
>
> **Q1: Hypernetwork's Sensitivity:** Thank you for raising this concern. To measure the hypernetwork's sensitivity to problem representations, we have conducted a targeted mismatched experiment. Specifically, we explicitly masked out the "node in sub-routes" flag in the multi-hot code when solving CVRP variants. The detailed results in **Table 1** show that even without the indicator, URS still achieves excellent zero-shot generalization performance, and a correct problem representation can further enhance performance. This not only demonstrates the robustness of URS but also highlights the utility of the hypernetwork.
>
> **Q2: Hypernetwork for STL:** Thank you for pointing this out. We have conducted a new ablation study comparing URS-STL with a variant that removes the hypernetwork (URS-STL w/o hypernet). The detailed results in **Table 2** show that employing the hypernetwork does not sacrifice STL performance. In contrast, the original URS-STL achieved superior performance on 9 out of the 11 evaluated seen variants.
>
> We will add all of the above results and analyses in the revised paper.

---

> > ### Author Rebuttal · Reviewer_STCX · 2026-04-01
> >
> > Thank the author(s) for their efforts. I believe the paper has met the quality standard for acceptance. I therefore maintain my rating of 5 (Accept).

---

> > > ### Author Response · Authors · 2026-04-02
> > >
> > > Thank you very much for your time and for your positive rating. We are glad to know your concerns have been addressed. We will carefully incorporate all your valuable suggestions into our final version. Thank you again for your constructive feedback and support!

---

### Official Review · Reviewer_VZ32 · 2026-03-13

**Soundness:** 2
**Presentation:** 3
**Significance:** 3
**Originality:** 3
**Overall Recommendation:** 3
**Confidence:** 3

**Summary:**

This paper presents URS, a unified neural routing solver designed to handle over 100 VRP variants using a single model without fine-tuning. The core innovation lies in the Unified Data Representation (UDR), which replaces discrete problem tags with a continuous feature manifold, effectively decoupling instance representation from constraint definitions. The architecture incorporates a Mixed Bias Module (MBM) to capture geometric and relational priors and a hypernetwork-based parameter generator that adapts decoder weights based on active constraint features. Evaluation covers 110 variants, including 99 unseen tasks, and demonstrates scalability to instances with up to 7,000 nodes.

**Compliance With Llm Reviewing Policy:**

Affirmed.

**Key Questions For Authors:**

Given that RouteFinder and CaDA are the closest concurrent works in cross-problem zero-shot routing, can you provide an empirical comparison on a subset of mutually supported VRP variants to validate your SOTA claims?

Regarding the LLM-driven mask generator in Appendix G, is there a formal verification or unit-testing protocol to ensure that the generated code never violates hard constraints like capacity or time windows?

What is the peak VRAM and latency penalty of using the MBM (triple-matrix concatenation) on a standard TSP compared to a standard multi-head attention mechanism?

How does the model handle edge cases where the Unified Attribute Set features are zero-filled, but the underlying problem logic still requires differentiation (e.g., distinguishing between a TSP and a CVRP with zero demand)?

**Limitations:**

The authors do not adequately address the failure modes of the LLM-driven mask generation, specifically the risk of generating subtly incorrect code for complex constraint combinations. Additionally, the discussion regarding the asymptotic optimality gap between URS and highly optimized heuristics like HGS-PyVRP is somewhat dismissive; the model's performance on the most challenging benchmarks suggests that neural constructive solvers still lag significantly behind specialized local search methods in absolute solution quality.

**Strengths And Weaknesses:**

The shift from explicit problem enumeration to a continuous Unified Data Representation is a theoretically elegant solution to the tag-collapse problem in multi-task NCO. The zero-shot generalization across a vast range of 110 variants is a significant empirical milestone for the field. The model demonstrates impressive robustness when scaling to 7,000-node instances, outperforming specialized baselines and classic heuristics on specific large-scale benchmarks.

The experimental validation is fundamentally compromised by the explicit exclusion of the most relevant concurrent foundation models, RouteFinder and CaDA. The authors admit on page 6 that these baselines were omitted due to "inconsistent problem settings," which is an unacceptable justification for a top-tier conference paper. RouteFinder and CaDA are the primary competitors in the "VRP Foundation Model" space; claiming state-of-the-art performance without a head-to-head zero-shot comparison on shared test distributions makes the empirical claims unsubstantiated. This omission is a critical flaw and suggests the evaluation may be cherry-picked to avoid stronger competition.

The reliance on an LLM-driven constraint satisfaction mechanism in Appendix G is theoretically unsound for hard combinatorial optimization. Using a probabilistic language model to generate executable masking functions introduces significant risk of hallucinations and logic errors in enforcing feasibility. For NP-hard problems where constraint satisfaction is a binary hard requirement, the lack of a formal verification step for these LLM-generated masks undermines the reliability of the solver in practical deployments.

The Mixed Bias Module (MBM) introduces substantial computational redundancy by statically concatenating distance, transposed distance, and relation matrices across all 12 attention layers for every problem instance. This leads to wasted channel capacity and increased VRAM usage during the evaluation of simple symmetric problems where asymmetric or relational priors are redundant. The paper fails to provide a thorough analysis of the trade-off between this architectural flexibility and its associated inference-time efficiency overhead.

---

> ### Author Rebuttal · Authors · 2026-03-31
>
> Thank you very much for your time and effort in reviewing our work. We provide an anonymous link (https://anonymous.4open.science/r/ICML26_Submission26157/Reviewer_VZ32.pdf) to present supplementary results.
>
> **W1/Q1: Comparison with RF and CaDA:** Thank you for pointing this out. In our initial manuscript, we strictly adhered to the problem settings established by MVMoE and ReLD to ensure a fair and consistent evaluation. Notably, as shown in **Table 6 of our manuscript**, URS already outperforms RF and CaDA on large-scale CVRPLib. In our cross-problem test, RF and CaDA are excluded due to discrepancies in the problems, including the number of trained CVRP variants (16 in their work vs. 5 in ours) and constraint settings (i.e., Backhauls and Time Windows).
>
> To further address your concern and ensure a fair comparison with RF and CaDA, we have conducted two additional experiments: (1) Evaluation on all shared seen and unseen VRP variants that have consistent problem definitions using their official models (see **Table 1**) and (2) retraining RF and CaDA under matched training settings with URS and evaluating the three models on 48 CVRP variants, the results in **Table 2** show that URS still achieves the best overall performance among the three methods and provides the best solutions for the majority of problems **(32/48)**, highlighting our strong zero-shot generalization.
>
> Additionally, to further validate URS's robustness, we have conducted a new empirical study using real-world CVRP instances and compared URS with mainstream generalist neural solvers (including RF and CaDA). We provide detailed results in **response to Q5 of Reviewer kR1Y**.
>
> **W2/Q2: Constraint Satisfaction:** We respectfully clarify a critical misunderstanding regarding our methodology. **URS does not utilize LLMs for constraint satisfaction at any point.** In Appendix G in our manuscript, we detail "Training and Model Settings" and contain no mention of LLMs.
>
> For constraint satisfaction, we have manually coded deterministic masking rules for each individual constraint (e.g., capacity). For VRP variants with multiple constraints, we apply the logical intersection of the relevant individual masks to filter the valid nodes at each decoding step. This deterministic approach guarantees 100\% formal feasibility.
>
> **W3: Effects of Deeper Encoder:** Thank you for raising this concern. We adopted a 12-layer encoder to enable the URS to better capture biases and obtain superior node embeddings. For symmetric problems, MBM enables URS to adaptively adjust distance biases from two distinct representational sub-spaces. If the relation matrix is absent in a problem, the model simply bypasses the dense attention computation, avoiding computational waste. To empirically substantiate this trade-off, we have conducted an ablation study comparing our 12-layer encoder against a 6-layer one (see **Table 3**). Since the heavy encoder is called only once during solution construction, there is no obvious time and memory difference between the models with 12- and 6-layer encoders. Thus, the performance gains heavily outweigh the minimal memory and time overhead.
>
> **Q3: Different Attention Mechanisms:** Thank you for raising this concern. We have conducted a new comparative experiment between different attentions and tested them on TSP instances to ensure consistent decoding steps. Specifically, we replace our MBM with both a standard MHA and the parallel MSA used in MatNet, and compare their complexity, memory overhead, and inference time (see **Table 4**). These results show that MBM imposes only modest computational penalties while successfully capturing crucial problem-specific biases, which enhances cross-problem generalization while reducing architectural redundancy.
>
> **Q4: Handle Edge Cases:** Thank you for raising this concern. Our UDR relies on the "node in sub-routes" binary feature in the Node-type Indicator to distinguish between a TSP and a CVRP with zero demand. The explicit structural role encoding within acts serves as a discriminator, preventing logic ambiguity and guiding correct routing behavior even when continuous attributes in the Unified Attribute Set are completely zero-filled.
>
> **Limitation: Comparison with HGS:** Thank you for raising this concern. HGS-PyVRP is a highly optimized improvement heuristic. In contrast, URS is an end-to-end constructive neural solver. Additionally, since URS is trained on small-scale synthetic instances with uniform distributions, the CVRPLib benchmark serves as an out-of-distribution generalization test for URS. Given these differences, we believe that the two solvers are incomparable. Notably, despite this paradigm difference, URS already shows impressive competitiveness. As shown in **Table 5 in our initial manuscript**, URS already outperforms HGS on large-scale CVRPB instances in less time.
>
> We will add all of the above results and analyses in the revised paper.

---

### Official Review · Reviewer_SSSM · 2026-03-18

**Soundness:** 2
**Presentation:** 2
**Significance:** 3
**Originality:** 3
**Overall Recommendation:** 5
**Confidence:** 3

**Summary:**

This paper proposes URS, a unified neural routing solver that aims to achieve zero-shot generalization across a wide range of unseen vehicle routing problem (VRP) variants with a single model. The method introduces a Unified Data Representation (UDR) to replace problem enumeration with data unification, a Mixed Bias Module (MBM) to capture multiple priors during encoding, and a problem-conditioned parameter generator to improve zero-shot generalization. Extensive experiments show that URS can handle 110 VRP variants, including 99 unseen variants, and scales to large instances with up to 7000 nodes. The paper positions itself as the first neural solver to handle over 100 VRP variants with a single model.

**Compliance With Llm Reviewing Policy:**

Affirmed.

**Final Justification:**

The author clearly distinguished the essential differences between UDR and existing methods such as GOAL, MVMoE in terms of feature decoupling and open constraint space, and clarified the innovation boundaries. Moreover, he supplemented the fairness comparison with RouteFinder and CaDA, improved the ablation experiments of the core components of UDR, enhanced the persuasiveness of the paper, and effectively answered my main questions regarding the experimental integrity. Overall, this rebuttal has strengthened our confidence in this method and its empirical verification.

**Key Questions For Authors:**

1. Can the authors clarify the essential differences between UDR and prior multi-task VRP methods such as GOAL and MVMoE, especially in terms of feature disentanglement and adaptation to open constraint spaces?

2. Can the authors provide more complete baseline comparisons, especially with RouteFinder and CaDA under matched settings, or otherwise give a more detailed justification for excluding them?

3. Can the authors provide more detailed ablation studies on the three core components of UDR, and quantify the computational complexity and parameter efficiency of MBM and the problem-conditioned parameter generator?

4. Can the authors analyze zero-shot generalization in a more stratified manner, especially for simple versus complex unseen constraint combinations and strongly coupled constraints such as asymmetric + pickup-and-delivery?

5. Can the authors further clarify the missing methodological details and ambiguous result presentation, such as the zero-padding rules in UDR, the role of (f(\alpha, N, d_{ij})), the meaning of negative Gap values, and the limitations on more complex VRP variants?

**Limitations:**

The paper should discuss its limitations more explicitly. In particular, it would be helpful to acknowledge the lack of stronger comparisons with mainstream baselines, the insufficient stratified analysis of zero-shot generalization, the incomplete ablations of UDR components, the limited robustness analysis on large-scale instances, and the lack of discussion on more complex VRP variants such as heterogeneous vehicles and dynamic VRP.

**Strengths And Weaknesses:**

**Core Contributions and Innovations: Need to Further Clarify the Boundaries of Innovation and Unique Value**

1. The core of the UDR proposed in the paper is to replace problem enumeration through data unification. However, in the existing multi-task VRP solutions, some work has already attempted to weaken the dependency on problem labels (such as GOAL and MVMoE). It is recommended to add more detailed comparative analysis in the Introduction/Method section to clarify the essential differences between UDR and existing methods in aspects such as feature disentanglement and adaptation to open constraint spaces, in order to avoid the innovation being obscured.

2. The Mixed Bias Module (MBM) integrates prior information from out-degree/in-degree distance matrices and relational matrices. The paper has verified its effectiveness, but it does not clarify MBM’s performance in terms of computational complexity. Compared with MatNet’s parallel layer design and traditional attention mechanisms, the parameter count and inference time increase/decrease ratio of MBM need to be quantified and supplemented to highlight the actual effect of "reducing architectural redundancy."

3. The Problem Condition Parameter Generator generates decoder parameters through a hypernetwork, replacing existing adapter fine-tuning schemes. It is recommended to supplement a comparison of the parameter counts between this generator and lightweight adapters, and analyze parameter efficiency on VRP instances of different scales, making the conclusion of "zero-shot generalization with no performance loss" more convincing.

**Related to Experimental Design and Analysis: Need to Supplement Key Comparisons, Ablation, and Robustness Verification**

1. **Completeness Issue in Baseline Method Selection:** The paper excludes RouteFinder and CaDA from the comparisons, merely stating "to maintain consistent training problem settings." However, these two methods are the mainstream baselines for current multi-task VRP solving. It is recommended to supplement comparison results with these two methods under the same training settings, or to provide a detailed justification for their exclusion (e.g., the training datasets and constraint sets are completely inconsistent) to avoid partiality in experimental comparisons.

2. **Insufficient In-Depth Analysis of Zero-Shot Generalization:** The paper verifies generalization performance on 99 unseen VRP variants but does not perform stratified analysis by the complexity of the constraints of the variants—for example, the change in URS's generalization gap under simple constraint combinations (single new constraint) versus complex constraint combinations (multiple new constraints stacked), and the bottleneck reasons for generalization performance under strongly coupled constraints such as asymmetric (A) + pickup and delivery (PD). Further investigation is needed.

3. **Need for Detailed Ablation Experiments:** The existing ablation experiments verify the synergistic effects of MBM, WEIGHT ((\lambda)), and BIAS ((\lambda)), but do not individually ablate the three core components of UDR (position identifier, unified attribute set, node type indicator), making it impossible to clarify each component's independent contribution to problem coverage and generalization ability. It is recommended to supplement ablation experiments for this part.

4. **Insufficient Robustness Verification on Large-Scale Instances:** The paper verifies scalability on 5000-7000 node instances but does not describe the constraint distribution characteristics of these large instances (e.g., whether they contain multiple depots, time windows, or other complex constraints), nor does it supplement convergence analysis of URS on large-scale instances (e.g., after training on 100-node instances, the performance degradation pattern at different scales). This should be improved.

5. **Insufficient Depth in Comparison with Classical Heuristic Solvers:** The paper compares with HGS-PyVRP on large-scale instances but does not compare LKH3 or OR-Tools on medium-scale instances (1000-3000 nodes), nor does it analyze URS's advantages in the "solution quality - runtime" trade-off—for example, in which scenarios URS can achieve similar quality with far lower runtime than classical solvers. It is recommended to supplement quantitative comparisons and analysis.

**Details and Presentation Related to Methodology: Need to Supplement Missing Details, Standardize Expressions, and Clarify Logic**

1. **Missing details in UDR's feature processing:** The paper mentions that UDR achieves attribute expansion/ablation through zero padding, but does not specify the zero padding rules for features in different VRP variants (e.g., the specific assignment of (\delta_i) when there are no demand features in TSP). It also does not explain how the sampling strategy of the random scalar (\eta_i) in the position identifier affects the solution of asymmetric problems. It is recommended to supplement this in Section 3.1 or the appendix.

2. **Ambiguous MBM computation details:** In MBM formulas (2)-(5), the specific form of the adaptation function (f(\alpha, N, d_{ij})) is only briefly mentioned in Appendix F. It is recommended to supplement the core role of this function in the main method section of 3.2 and explain why the scale (N) is removed from the relation matrix (R), making the methodology description more self-consistent.

3. **Insufficient rationale analysis for training strategy:** The paper selects 11 training problems following the "minimal redundancy principle," but does not quantify the feature coverage of the training problems (e.g., the occurrence frequency of each UDR feature in the training set). It also does not verify whether the conclusion that "Penalty attributes appearing in a single problem are sufficient for generalization" is universal (e.g., how the generalization performance would be for other attributes if they appear only once). It is recommended to supplement these analyses.

4. **Contradictory/ambiguous expression of some experimental results:** For example, some Gap values for URS in Table 2 are higher than URS-STL. The paper only states that "mixed training results in a slight performance drop" but does not analyze which problems show more significant performance degradation and why. In Table 4, URS shows negative Gap values for some unseen variants; it is necessary to clarify the definition of negative Gap (whether it indicates outperforming the Oracle) to avoid misunderstanding by readers.

5. **Overly brief analysis of limitations:** The paper points out that the current training scheme does not consider the complexity differences among VRP variants, but does not propose specific improvement directions (e.g., hierarchical training, difficulty-adaptive sampling). It also does not analyze the generalization potential and bottlenecks of URS for more complex VRP variants (e.g., heterogeneous vehicles, dynamic VRP). It is recommended to enrich the analysis of limitations and future work.

---

> ### Author Rebuttal · Authors · 2026-03-31
>
> Thank you very much for your time and effort in reviewing our work. We provide an anonymous link (https://anonymous.4open.science/r/ICML26_Submission26157/Reviewer_SSSM.pdf) to present supplementary results.
>
> **1. Differences between UDR and Prior Works:** Thank you for pointing this out. Our UDR decouples diverse problem instances from constraints and operates in a border space with both continuous and discrete features, providing a more universal data representation. By representing open constraint spaces via specific feature slots, we enable zero-shot generalization to nearly 100 variants.
>
> From a data perspective, the inputs to GOAL (problem-specific inputs) and MVMoE (strictly adhering to CVRPTW) can be viewed as special subsets of UDR. We will add the above analysis in the Method section of the revised paper.
>
> **2. MBM vs. Existing Attentions:** Thank you for raising this concern. For a comparison between MBM and existing attentions (i.e., MHA and MSA in MatNet), please refer to the response to **Q3 of Reviewer VZ32**.
>
> **3. Hypernetwork vs. Adapters:** Thank you for raising this concern. We have trained URS-Adapter, a new URS variant that adopting a distinct decoder for each of the 11 variants, and compare it with URS-Hypernet on 11 seen problems (see **Table 1**) and complex CVRP variants of different scales (see **Table 2**).
>
> **4. Comparison with RF and CaDA:** Thank you for raising this concern. For a comprehensive comparison with RF and CaDA, please refer to the response to **W1 of Reviewer VZ32**.
>
> **5. Stratified Analysis:** Following your suggestion, we have conducted a stratified analysis grouping VRP variants by their constraint complexity and focused on combinations of C, O, PD, and A constraints (**Figure 1**). For the A and PD combination, we attribute the bottleneck to the sparsity of signals during training: the shared MBM encoder predominantly observes symmetric spatial patterns, and relational attention is activated only once (PDTSP). It is challenging for URS to simultaneously navigate complex directional distance biases and strict relational precedence. Handling such complex constraint combinations is an important research topic for our future work.
>
> **6. Ablation Study of UDR:** Thank you for raising this concern. We have conducted a new ablation study to clarify the independent contribution of each UDR component (see **Table 3**).
>
> **7. Robustness Verification:** Thank you for pointing this out. We will address each of your concerns one by one: (1) For instances with more than 5K nodes, we use CVRPLib XXL benchmark, which contains only capacity constraints; (2) We have included OR-Tools for cross-problem scalability (see **Table 4**). LKH3 is excluded because it produces infeasible solutions in these instances; (3) We have plotted the optimality gap trends for URS and neural baselines on large-scale instances (**Figure 2**); (4) For the "quality-time" trade-off, URS offers a clear advantage in many variants, like large-scale CVRPB. This efficiency advantage extends to our zero-shot generalization on multiple unseen variants (see **Table 5**).
>
> **8. Zero Padding and Random Scalar:** Thank you for raising this concern. For zero-padding rules, if a feature does not exist in a specific VRP (e.g., demand in TSP), we simply set it to 0. Because asymmetric VRPs lack coordinates, we introduce a random scalar to ensure that each node has a distinct input. We have conducted an ablation study by removing $\eta_i$ and evaluating the model across 16 common AVRP variants (see **Table 6**).
>
> **9. Adaptation Function:** Thank you for pointing this out. The adaptation function (f($\alpha$, $N$, $d_{ij}$)) is designed to enhance the model's ability to adaptively capture problem-specific biases across different scales. Because the pickup-delivery pairing is a logical constraint and fundamentally scale-independent, we remove $N$ from the relation matrix.
>
> **10. Training Problem Selection:** Thank you for pointing this out. For detailed results and analysis, please see the **response to W2 of Reviewer kR1Y**.
>
> **11. Ambiguous expression and Negative Gap:** Thank you for pointing this out. (1) We will revise our expression and provide a detailed analysis in the revised paper. Taking ATSP as an example, we attribute the URS-MTL's performance degradation to an inequality in geometric structures, which likely shifts the learning capacity to symmetric VRPs. (2) The negative gap indicates that URS produces a better solution than Oracle. Additionally, to prevent any reader confusion, we will carefully review the manuscript and revise any inappropriate or ambiguous expressions.
>
> **12. Analyses of Limitations:** Thank you for these constructive suggestions. In the revised paper, we will enrich the analysis of limitations and future work, including specific directions for improving training and applicability to more complex variants.
>
> We will add all of the above results and analyses in the revised paper.

---

### Decision · Program_Chairs · 2026-04-30

**Decision:**

Accept (regular)

**Comment:**

This paper proposes URS, a unified neural routing solver targeting cross-problem zero-shot generalization across a large number of VRP variants. All reviewers acknowledge the ambition and potential impact of this work, particularly the ability to handle over 100 variants with a single model and the introduction of a unified data representation (UDR).

Summary of Reviews.
Two reviewers (STCX, SSSM) provide positive assessments, highlighting the novelty of UDR and the strong empirical performance, especially after rebuttal clarifications. Two reviewers (VZ32, kR1Y) raise concerns regarding experimental completeness (notably comparisons with RouteFinder/CaDA), efficiency analysis, and the trade-off between generality and performance. VZ32 maintains a weak reject mainly due to missing baseline comparisons, while kR1Y questions training design and performance trade-offs.

Discussion and Rebuttal.
The authors provide a thorough and detailed rebuttal. In particular:

The concern about missing comparisons with RouteFinder and CaDA is substantially addressed through additional experiments under matched settings, showing competitive or superior performance.
A critical misunderstanding regarding LLM-based constraint handling is clearly clarified; the method relies on deterministic masking, resolving a major soundness concern.
Additional ablations, stratified generalization analysis, and clarifications on methodology and experimental details improve the completeness and transparency of the work.

While some concerns remain (e.g., efficiency analysis and deeper understanding of generalization boundaries), they are relatively minor and do not undermine the core contributions.

Final Assessment.
Overall, the paper presents a significant step forward in neural combinatorial optimization, particularly in scaling zero-shot generalization across diverse VRP variants. The rebuttal successfully addresses the most critical concerns, and the remaining issues are suitable for revision rather than rejection.